# Understanding Low Vaccine Uptake in the Context of Public Health in High-Income Countries: A Scoping Review

**DOI:** 10.3390/vaccines12030269

**Published:** 2024-03-04

**Authors:** Josephine Etowa, Sheryl Beauchamp, Manal Fseifes, Glory Osandatuwa, Paul Brenneman, Kudirat Salam-Alada, Rasheedaht Sulaiman, Emmanuella Okolie, Ihechi Dinneh, Samora Julmisse, Victoria Cole

**Affiliations:** 1School of Nursing, Faculty of Health Sciences, University of Ottawa, Ottawa, ON K1S 5S9, Canada; sbeau144@uottawa.ca (S.B.); mfsei072@uottawa.ca (M.F.); paulbrenneman@cmail.carleton.ca (P.B.); ksala063@uottawa.ca (K.S.-A.);; 2Faculty of Health Science, University of Ottawa, Ottawa, ON K1N 6N5, Canada; vcole@uottawa.ca

**Keywords:** vaccine hesitancy, vaccine uptake, COVID-19 vaccination, African, Black and Caribbean, high-income countries

## Abstract

Although the COVID-19 pandemic has caused the need for the largest mass vaccination campaign ever undertaken to date, African, Caribbean, and Black (ACB) populations have shown both a disproportionately high degree of negative impacts from the pandemic and the lowest willingness to become vaccinated. This scoping review aims to investigate low vaccine uptake in ACB populations relative to public health in high-income countries. A search was conducted in MEDLINE(R) ALL (OvidSP), Embase (OvidSP), CINAHL (EBSCOHost), APA PsycInfo (OvidSP), the Cochrane Central Register of Controlled Trials (OvidSP), the Cochrane Database of Systematic Reviews (OvidSP), the Allied and Complimentary Medicine Database (Ovid SP), and the Web of Science following the Joanna Briggs Institute (JBI) framework for scoping reviews, supplemented by PRISMA-ScR. Theoretical underpinnings of the intersectionality approach were also used to help interpret the complexities of health inequities in the ACB population. The eligibility criteria were based on the population, concept, context (PCC) framework, and publications from 2020–19 July 2022 which discussed vaccine uptake amongst ACB people in high-income countries were included. Analysis was carried out through thematic mapping and produced four main themes: (1) racism and inequities, (2) sentiments and behaviors, (3) knowledge and communication, and (4) engagement and influence. This study has contributed to the identification and definition of the issue of low vaccine uptake in ACB populations and has illustrated the complexity of the problems, as vaccine access is hampered by knowledge, psychological, socioeconomic, and organizational barriers at the individual, organizational, and systemic levels, leading to structural inequities that have manifested as low vaccine uptake.

## 1. Introduction

African, Caribbean, and Black (ACB) populations are not only vulnerable due to health inequities, as evidenced by higher rates of SARS-CoV-2 infections, hospitalizations, and associated mortalities, but are also the least willing to receive the coronavirus disease of 2019 (COVID-19) vaccine [1,2,3]. The COVID-19 pandemic has been one of the greatest public health threats of modern times, bringing societal, community, and individual challenges to the forefront; these impacts have been intensified in racialized communities as pre-existing inequities and vulnerabilities are exacerbated [4,5]. These inequities are strongly influenced by socioeconomic factors, referred to as the social determinants of health (SDOH); for example, death rates in ACB populations were higher in areas with a greater incidence of adverse SDOH [5,6]. Although race-based data collection remains inconsistent in Canada, the cities of Ottawa and Toronto reported 1.5-5 times the increase in COVID-19 infection rates among racialized communities; these findings are consistent with other high-income countries, including the United States (US) and the United Kingdom (UK) [5,7,8]. These highlighted COVID-19 racial inequities regarding disease and vaccinations in ACB populations are not new; public health disparities also occurred during the 2009 H1N1 pandemic, and high rates of vaccination mistrust have been reported for the Human Papillomavirus (HPV) vaccine, H1N1 vaccine, and influenza vaccine [4,9,10,11,12].

The World Health Organization (WHO) [13] examined the refusal or reluctance to become vaccinated despite available vaccines; reasons identified include complacency, inconvenience, and lack of confidence. It can result in delayed vaccination or uncertainty in the vaccine even after its administration, which can threaten vaccination programs by leading to decreased coverage and increased risk of vaccine-preventable disease outbreaks [14,15]. Currently, vaccine hesitancy regarding the COVID-19 vaccination threatens the success of the most extensive mass vaccination campaign ever undertaken, both within Canada and globally [16]. However, the importance of addressing vaccine hesitancy reaches beyond its implications for COVID-19, including both current and potential future outbreaks; the coronavirus alone has accounted for three pandemics within the last 20 years, including COVID-19, severe acute respiratory syndrome (SARS) and Middle East respiratory syndrome (MERS) [17]. 

Vaccine uptake has been declining over the past several decades; global coverage dropped from 86% in 2019 to 83% in 2020, with the highest rate of children under one, 23 million, not receiving essential vaccines since 2009, and completely unvaccinated children increasing by 3.4 million [18,19]. In 2019, the World Health Organization (WHO) listed vaccine hesitancy and weak primary healthcare as two of the top ten threats to global health; both threaten the success of vaccination campaigns [13,20]. 

In Canada and globally, vaccines significantly prevent and control infectious diseases and are thus a cornerstone of public health [21]. Historically vaccines have reduced disease-specific mortality rates including smallpox, rabies, polio, the plague, typhoid and many more, and have significantly decreased infant mortality rates globally [18]; over 3 million child deaths are estimated to be prevented each year globally through vaccinations [22]. Despite vaccinations being considered to be one of the public health’s greatest success stories, vaccine hesitancy is influenced by the confidence in the competencies of health professionals and health services [23,24]. Vigilance is required to maintain and increase vaccine uptake, especially in vulnerable populations. People’s behaviors and willingness to follow recommended measures are the most powerful tools against the viral spread [25,26]. 

Public health interventions must go beyond COVID-19 and seek to understand the historical basis for vaccine hesitancy while adapting to the current dynamics and preparing for future outbreaks. If public health fails to implement appropriate interventions, health inequities threaten to become even more vast, as those who are socioeconomically disadvantaged often have health conditions that are exacerbated by inadequate healthcare [12,27,28]. Further research is needed to determine why ACB populations have the lowest level of vaccine acceptance in Canada and other high-income countries [3,29]. These disparities need to be promptly addressed; however, a greater understanding of the implications of challenges faced by vulnerable populations on vaccine uptake and public health is required [12,28]. 

Before embarking on creating service provider (i.e., healthcare providers, policymakers, community organization providers) interventions applicable to vaccine hesitancy in vulnerable ACB populations, associated concepts and their boundaries must be clarified. Due to the explorative nature and broad overview desired, a scoping review (ScR) approach has been chosen. There has been a steady increase in the use of ScRs, as they are valuable for health researchers to establish the breadth of data available [30,31]. A ScR does not undergo an assessment of bias, including critically appraising the evidence sources, so the implication for service providers would be better served through the SR; however, if the evidence sources reveal any potential implication to service provider practice, service provider knowledge and research, these will be stated [32,33]. The essential characteristics of this ScR will include pre-planning through the creation of a protocol, transparency of the processes involved, and clarity of concepts [31].

A preliminary search of existing scoping reviews, systematic reviews and protocols was performed on 31 January 2022, relating to the determinants of vaccine hesitancy and ACB populations in the JBI Evidence Synthesis, Cochrane Database of Systematic Reviews, Cumulative Index to Nursing and Allied Health Literature (CINAHL) and PubMed. Several terms were piloted (refer to Appendix B for the search strategy), and the keywords “vaccine hesitancy “ and “Black” yielded the most relevant results, namely, the following four reviews: One rapid systematic review related to COVID-19 vaccine hesitancy and minority ethnic groups in the UK [34]; one scoping review related to COVID-19 vaccine hesitancy globally [35]; and two systematic reviews, one related to vaccine hesitancy in the US [36] and the second related to vaccine acceptance in different populations in the US [37]. In addition, on 22 June 2022, VC searched the Open Science Framework (OFS) site and found the ScR protocol on racial and ethnic minorities and Indigenous Population groups living in high-income countries was found in the OFS [38]. To avoid duplication of findings this ScR included all vaccines, ACB populations specifically, all high-income countries where the ACB population is considered a minority population, and evidence sources from 2020-2022. 

To develop a clear study structure and to help guide the selection of evidence sources that align with the research question, the Joanna Briggs Institute (JBI) recommended PCC (population, concepts, context) framework was used; PCC was also used to compose the title as per the JBI framework [32]. The research question is: What are the determinants and interventions of low vaccine uptake in African, Caribbean, and Black (ACB) populations relative to healthcare in high-income countries? Whereby the population (P) is ACB populations and service providers, concept (C) is low vaccine uptake, and the context (C) is healthcare in high income countries. It is imperative to note that there is a dearth of research within the literature that adequately address the challenges that ACB populations face with vaccination programs. While this scoping review include all high-income countries, the articles that met the inclusion criteria were mostly from the US, Canada and the UK. Given the lack of ACB focused literature from other high-income countries this review uses the available sources to illustrate existing models that can be utilized within our healthcare system and to illuminate the need for all high-income countries to collect and use race-base data for the development of targeted interventions.

The main aim of this ScR is to explore low vaccine uptake in ACB populations relative to public health in high income countries. The objectives are (1) To identify concepts and boundaries of existing evidence sources on low vaccine uptake in ACB populations; (2) To map the evidence on the concepts and boundaries and to identify gaps in the research, and (3) To determine existing interventions to improve low vaccine uptake in the study population. These objectives were achieved through systematically reviewing the breadth and types of source evidence available; the software program Covidence was used to manage and select source evidence, which was open to all types of evidence sources, and data extraction from source information was done manually. The data identified ways that public health addresses or fails to address vaccine uptake, these were reported in the findings through descriptive narrative. This ScR will describe the methodology, including the search strategies and selection of evidence sources, the analysis and presentation of the findings, and a discussion and conclusion. 

## 2. Materials and Methods

Intersectionality, the socioeconomic model (SEM), and the social determinants of health (SDOH) approach were used to inform the analysis and the interpretation related to the complexities of low vaccine uptake in ACB populations. Intersectionality highlights the unique forms of discrimination faced by ACB and other vulnerable populations. Through this perspective, it is recognized that inequities result from multiple factors related to the intersection of power relation, experiences, and social locations [39]; the intersection of these factors exacerbates health inequities [40]. In addition, the socioeconomic model (SEM) describes how characteristics of the environment and the individual interact at multiple levels (macro/systems, meso/organizational, and micro/individual) to influence health outcomes [41]. Furthermore, the social determinants of health (SDOH) describe factors that drive health outcomes and influence health-related behaviours [42]; in Canada, they are listed as income and social status, employment and working condition, education and literacy, childhood experiences, physical environments, social supports and coping skills, healthy behaviours, access to health services, biology and genetic endowment, gender, culture and race/racism [43]. 

The Joanna Briggs Institute (JBI) scoping review (ScR) methodological framework by Peters et al. [32]. was used to help provide organization to this ScR, which aimed to outline different types of evidence on the determinants and interventions of low vaccine uptake in racialized and marginalized populations, as well as the gaps for future research. The nine steps of the framework are; (1) defining and aligning the objectives and research question; (2) developing and aligning the inclusion criteria with the objectives and question; (3) describing the planned approach to evidence searching, selection, data extraction and evidence presentation; (4) searching the evidence; (5) selecting the evidence; (6) extracting the evidence; (7) analyzing the evidence; (8) presenting the evidence; and (9) summarizing the evidence about the purpose of the review, making conclusions, and noting any implications of the findings [32].

### 2.1. Defining and Aligning the Objectives and Question

Objective (1) To identify concepts and boundaries of existing evidence sources on low vaccine uptake in ACB populations, which relates to the research question by determining how the concept of low vaccine uptake is identified and defined by current literature. Objective (2) To map the evidence on the concepts and boundaries and to identify gaps in the research, relates to the research question by determining links between the concepts to elaborate on the complexities of low vaccine uptake by ACB populations, as well as gaps in the literature that may require further research, and (3) To determine existing interventions to improve low vaccine uptake in the study population, relates to the research question by expanding on what interventions have been attempted or implemented to improve low vaccine uptake, this can also help to identify gaps for future research, or provide information to create new interventions. 

### 2.2. Developing and Aligning the Inclusion Criteria with the Objectives and Question

The eligibility criteria with explanation in PCC framework: 

The population (P) includes any person, community, or population that is identified as ACB, or as a service provider for this group.

The ACB populations were chosen because of their significant levels of vulnerability and high propensity for low vaccine uptake. The service providers were selected based on experiences regarding determinants and interventions related to vaccine uptake.

The concept (C) of low vaccine uptake encompasses reasons about sentiments for not getting vaccinated as well as vaccine accessibility issues. Due to the increasing trend of low vaccine uptake, measures should be taken to identify and understand underlying factors, and to inform the creation of effective and targeted solutions [18,19,44]. In this ScR, attempts will be made to explore all relevant concepts and their boundaries, with the dates of publication restricted to 2021 to 19 July 2022, to provide contemporary evidence sources that are also inclusive to the public availability of the COVID-19 vaccine [3,18,19,45]. 

The context (C) of healthcare will include all settings, such as public health institutions, hospitals, clinics, and so on, without limitation. The evidence source must describe this study population in a high-income country as defined by the World Bank (Appendix C), where ACB populations are considered a minority. This is to aid in exploring the roles of race and ethnicity on the health of ACB populations in Canada, which has often been extrapolated from high-income countries, such as by using US- and UK-derived statistics, two countries that systematically collect race-based health data, whereas collecting race-based data in Canada can be challenging [8,46,47]. In addition, high-income countries are more likely to have the financial means to provide equitable care.

The ambiguity of some of the terms used within the research questions are consistent with [48], which recommends maintaining a broad approach to generate a breadth of literature; more parameters can be added once a general scope and sense of volume are obtained. Furthermore, only English and French evidence sources were considered due to feasibility issues about the lack of available interpreters and the research team members being fluent in English and/or French; this is consistent with JBI, which states that no restrictions on language are recommended, but if not feasible this should be clearly stated [32].

Therefore, based on the PCC framework, the inclusion criteria for this study was African, Caribbean, and Black populations, and other related terms; high-income countries as defined by the World Bank (Appendix C) where ACB populations are considered a minority; all service providers, English and French languages; open to all types of evidence; related to low vaccine uptake and alternative terms; all vaccines; and publications in 2020 to 19 July 2022. The exclusion criteria were any evidence sources that did not meet the inclusion criteria.

### 2.3. Planned Approach to Evidence Searching, Selection, Data Extraction and Evidence

Refer to protocol [49].

### 2.4. Searching the Evidence 

A comprehensive search strategy was developed to identify relevant evidence sources based on the research question and PCC framework. To increase the breadth of the ScR, source evidence was open to all types, such as primary studies, secondary studies, poster presentations, abstracts, conference proceedings, commentaries and reports. In addition, only evidence sources with a primary focus on African, Caribbean, and/or Black populations were included so that issues that are specific to this target population were addressed. Furthermore, the timeframe will be restricted to 2020–19 July 2022 to provide contemporary findings given the dynamic nature of vaccine uptake, particularly during the COVID-19 pandemic. All evidence sources from the searches were recorded in the Preferred Reporting Items for Systematic Reviews (PRISMA) flowchart adapted from Page et al. [50]. (Figure 1). 

The search strategy followed the three-step process recommended by JBI [32]; step 1, involved an initial search in identifying a list of terms; step 2 implemented the search strategy based on the identified terms; and step 3 involved the search of the references from selected articles, no further hand searchers or directly contacting authors was performed due to time limitations. 

#### 2.4.1. Step 1: Initial Search to Identify the List of Relevant Terms

To help capture the breadth of low vaccine uptake in ACB populations relative to public health in high-income countries, a piloting of synonyms, variations, and associative terms was performed, with the implementation of keywords and Medical Subject Headings (MeSH) terms being reviewed by research team members SB and JE with the information specialist VC in June 2022. Initial searches were used to inform the iterative process of the ScR, by potentially refining and allowing for new sources and keywords to be added [32]. 

#### 2.4.2. Step 2: Implementation of Search Strategy Based on Identified Terms 

Search strategies were developed by VC (an information specialist) and peer-reviewed using the PRESS guideline [51]. The search was conducted in the following databases: MEDLINE(R) ALL (OvidSP), Embase (OvidSP), CINAHL (EBSCOHost), APA PsycInfo (OvidSP), the Cochrane Central Register of Controlled Trials (OvidSP), the Cochrane Database of Systematic Reviews (OvidSP), the Allied and Complimentary Medicine Database (Ovid SP), and the Web of Science. Each database was searched from its inception until 19 July 2022, for the concept of “Vaccine uptake” and “African, Black and Caribbean” populations using a combination of subject headings and keywords. 

Drafting the search strategy was informed by two Cochrane reviews, Abdullahi et al. [52], and Cooper et al. [53], and a protocol in Open Science Framework by Thota [38], was used to inform the concept of vaccine uptake. The concept of African, Caribbean, and Black was informed by consulting Hope, et al. [54] systematic review. No search filters or language limits were used, and no publication restrictions were applied to the search. The search strategies of the databases can be found in Appendix B.

#### 2.4.3. Step 3: Hand Searches and Reference List

A manual search of the reference lists of the retrieved articles was conducted to capture all relevant evidence sources for potential inclusion by research team members (MA, PB, GO, KS, RS, ECO, ID, SJ, HO, SB, JE). Any reference that met all the eligibility criteria was reviewed for duplication against our initial list of articles; any duplicates were removed. Due to time limitation, further hand searches were not performed. 

### 2.5. Selecting the Evidence

All evidence sources that resulted from the database searches were entered into the software program Covidence (Veritas Health Information, Melbourne, Australia). They were then screened first by their title and abstract and afterwards by their full-texts based on the aforementioned eligibility criteria by two members of the research team (MA, PB, GO, RS, ECO, ID, SJ, SB and JE) and a third team member resolved any conflicts. The numbers of excluded articles were recorded in the PRISMA flowchart (Figure 1) with the rationale for their exclusion. The initial search of the 8 databases yielded a total of 9378 articles, and the program Covidence removed 4246 duplicates. Subsequently, 5132 articles were screened independently by 2 members of the research team (as described above), and a further 2746 were excluded. The remaining 2386 articles had their full text reviewed for relevance to the research question while meeting all the eligibility criteria; of these, 60 articles remained; reasons for exclusion include wrong time range, population, setting, design, indication, intervention, outcome, no population mentioned and duplicates. No new articles were added from the references; those references that met the selection criteria were duplicates of previously selected articles and were thus not included. The 60 articles that met all the criteria underwent the data extraction process. There were 10 qualitative, 24 quantitative, 7 mixed methods, and 19 commentary articles. 

## 3. Results

### 3.1. Extracting the Evidence 

To create a descriptive narrative of the evidence sources, an extraction table was created to provide an analytical framework [48]. The data was analyzed using Thematic Mapping [49]. Before data extraction, the articles were separated by article type; qualitative, quantitative, mixed-methods, and commentaries were the article types found in the evidence sources for this ScR. 

Creating the extraction table was an iterative process [55]. Consistent with Peters et al. [32], the table was calibrated in our study by the team members (PB, GO, KS, RS, ECO, ID, SJ, SB, JE) by independently extracting the same three articles; the results were subsequently compared, and any discrepancies were discussed until a consensus was reached. The resultant table contained general information and information that can help inform the research questions and objectives, namely, first author, date, title, country of origin, population, type of vaccine, study design, sample size, relevant findings, and conclusions.

Data extraction was performed by research team members (MA, PB, GO, KS, RS, ECO, SJ, SB and JE), and two team members independently extracted each article. The extracted data from each evidence source aligned with the research question and objectives and was used to inform the collation and summarization of the findings from the evidence sources. Refer to Table 1 for the qualitative extraction table, Table 2 for the quantitative extraction table, Table 3 for the mixed methods extraction table, and Table 4 for the commentary extraction table.

#### 3.1.1. Qualitative Article Extraction

**Table 1 vaccines-12-00269-t001:** Data extracted from qualitative articles. The first author and date of publication, country, population, vaccine, design sample, size, findings, and conclusion relevant to the research question for each of the evidence sources.

	Article	Country	Population	Vaccine	Design	Sample Size	Findings	Conclusion
1	[56]	USA	Black adolescents	COVID-19	Qualitative, in-depth interview (IDI)	28	- Behaviors and attitudes of church officials and older family members, misinformation from the Internet and peers, personal fears, and skepticism towards the healthcare system and government influenced likelihood of vaccine acceptance	- Tailored messaging to reduce vaccine-related skepticism and address misinformation related to side effects and governmental distrust - Older family members and church officials have the social capital promote vaccination
2	[57]	USA	Black communities	COVID-19	Qualitative, rapid review	61 articles	- Promote vaccine uptake by addressing mistrust, misinformation, and improving access by using trusted communication channels, address historic and experience-based reasons, hold town halls, culturally competent outreach, Black physicians and clinicians partner with community leaders, and trusted and convenient vaccination sites	- HCP * should link with the community through outreach, social media and partnering with community leaders - Distrust is a well-founded response to structural racism; structural inequities and racism are the problem
3	[58]	USA	Black patients	COVID-19	Qualitative semi-structured interview (supplemental)	37	- Higher prevalence of mistrust about vaccine efficacy, safety, and equitable distribution of the COVID-19 vaccine; PPE and staying home more effective - Fear of racial discrimination of treatment and intended to wait until others received the vaccine first	- Decisions based on discussions with their clinicians and observations of vaccine rollout. - Awareness of historical distrust and the acceptance of new medical intervention can inform efforts to empower Black patients
4	[59]	USA	-Black Americans: expressed low vaccine intentions -Stakeholders: communities highly impacted by COVID-19	COVID-19	Qualitative semi-structured interview	-24 Black Americans -5 Stakeholders	- A “wait-and-see” approach, for side effects and efficacy - Systemic racism: perceived barriers of structural, technology, transportation, medical mistrust of vaccines, healthcare providers, government, health systems, and pharmaceutical companies - Vaccine promotion: strategies acknowledge systemic racism as the root of mistrust, preferred and transparent messaging about side effects, non-medical and medical sites, trusted sources of information, such as Black doctors, researchers, and trusted leaders - Mistrust in providers and the health system: provider’s lack of cultural sensitivity, responsiveness, and competency in practice	- Campaigns: open dialogues with trusted and credible scientists and HCPs *, streamline and maximize process, acknowledge and address mistrust to increase equitable vaccine access by improving confidence and intentions
5	[60]	USA	AA (African American) parents	HPV (human papillomavirus vaccine)	Qualitative, focus group discussion (FDG) with demographic survey	18	- Wanting to be informed, concerns of unfamiliarity, mistrust of HCPs *, pharmaceutical companies, and the government, clarifying risk/benefits, cancer prevention, using straightforward language, provider recommendation - Effective messaging strategies: visuals and narratives with diversity across race, age and gender, clear language on eligibility, transparency on side effects, additional sources of information - Message dissemination: physical locations, word of mouth, and social media	- Promotions: tailored to AA parents and their children to consider building trust and representation in promotional materials - Highlights the importance of culturally tailored messaging with the faces and voices of the intended AA audience to build trust in the community
6	[61]	USA	AA adults	COVID-19	Qualitative, IDI	21	- VH * determinants: historical mistrust due to government and pharmaceutical companies, unethical research in Black populations, and continued acts of violence by police are a source of tension and distrust of government, knowledge and awareness of the vaccine, social media misinformation, perceptions of HCP *, concerns of side effects, the newness of the vaccine, its necessity and safety, political affiliation -Infodemic exacerbated poor health literacy - Negative experiences with HCP * worsened VH * and validated distrust - Community-based and faith-based health and wellness programs were trusted information sources, enrolled community members for vaccination, organized vaccine clinics at AA churches, and connected community members to HCPs *	- Government needs to commit resources to addressing historical factors and building trust - Partnerships with community members, church leaders and local government to increase community capacity by co-creating solutions including PH * messaging to increase trust and vaccine uptake - Strategies: address relationship with police, increasing communication and collaboration between HCP *, AA, and the government, and government advocating for programs and policies to elevate AA communities
7	[62]	USA	Black Americans	COVID-19	Qualitative, FGD	24 4 focus groups	- Mistrust for the scientific research organizations, medical establishments and pharmaceutical companies based on historical unethical mistreatment, quick development of the vaccine, political environment promoting racial injustice, limited data on short- and long-term effects were reasons for VH *, wrong approach as efforts should be on improving baseline health, confidence lowered by conflicting guidance from the federal, state, and local governments as well as political meddling - Those with VH * expected extremely low vaccine uptake in their social network and a famous Black person would not influence their decision - Increased willingness: safety, efficacy, adverse side effects, transparency in its development, protect families and small children, safely return to work, believed the vaccine would not harm them, reassurance, and recommendations from trusted HCP *; negativity may influence against vaccination - Health concerns: infected by vaccine, risk to immune system with comorbidities	- Build trusted client–HCP * relationships, educate, recommend COVID-19 to those who are VH * - Identify Black influencer to to advocate for the vaccine
8	[63]	USA	Black Americans	COVID-19	Qualitative, systematic review	26 articles	- Mistrust in the government and healthcare systems; 93% of articles had concerns mistrust and VH * - Concerns related to safety, side effect and misconception with COVID-19 vaccine raises ethical questions about health literacy levels and how to it can be improved in Black communities - Unfair distribution of research burdens and benefits can result when Black individuals are represented in the research as a generalization to ethnic minority groups. - Patient–provider relationships can help build trust and reduce skepticism - Community programs that are transparent and factual with officials that have an established relationship with community leaders and that follows up with community members to meet their needs can help to rebuild relationships with Black communities and government and PH * officials	- Healthcare providers and agencies should ethically consider the current drivers and effects of mistrust, adequate inclusion of vulnerable populations in research, improvement of health literacy, and the role of physician in the heath of Black Americans
9	[64]	USA	Black mothers	All	Qualitative review, blogs, social media posts, and comments that reflect vaccine-critical sentiments, mostly Twitter and Facebook	249 threads with 311 posts	- Black mothers experience gender and racial bias and rejection of vaccines is a form of governmental power resistance - Black mothers are concerned about vaccines and the organization that promote them and have considered homeschooling as a legal way to avoid vaccines; warn others not to state their objections as they could be more vulnerable, such as to child protective services and to a loss of benefits - Mandatory vaccines in children’s programs may alienate families	- Rather than structural barriers, the under-vaccination can be due to intentional refusal and parental agency - VH * Black mothers view physicians as a potential threat to report families to state agencies; not as consultants or service providers the way privileged families do - Many Black mothers question entities that produce, market, and distribute vaccines - Distrust has been increased by the lack of access to healthcare and vaccines related to COVID-19
10	[65]	USA	AA community leaders	COVID-19	Qualitative, FGD	18	- Gaining trust is essential for health communication; trusted messengers are important for disseminating accurate information and promoting vaccination behaviors in AA communities - Messengers, such as student leaders, coaches, and faculty can deliver vaccine-related messages to AA students - Those who obtained their information from less reliable sources had a higher likelihood of misinformation, which led to higher levels of VH *, whereas those who obtained their information from physicians and professionals had a better understanding - Receiving incentives, such as payment, for vaccination caused suspicion - Community leaders recommended recruiting trusted messengers, using football games, homecoming events, and other social events to reach target populations as well as conducting health communication campaigns with open dialogue among stakeholders	- Misinformation and mistrust are the main drivers of VH * in AA communities - Vaccine promotion should include trust building activities, transparency about vaccine development, and community engagement - Interventions regarding communication, accurate messaging, and behavioral change need community support and engagement - To increase trust and confidence, have a combination of key messengers, social events, and use multi-source social media - Tailoring messaging for certain groups, such as by age, may reduce misinformation and promote vaccination in the community

* VH = vaccine hesitancy; HCP = healthcare provider; SDOH = social determinants of health; PH = public health.

#### 3.1.2. Quantitative Article Data Extraction

**Table 2 vaccines-12-00269-t002:** Data extracted from quantitative articles. The first author and date of publication, country, population, vaccine, design sample, size, findings, and conclusion relevant to the research question for each of the evidence sources.

Number	Article	Country	Population	Vaccine	Design	Sample Size	Findings	Conclusion
1	[66]	USA	Black people	COVID-19	Quantitative abstract - 3-tiered approach to improve vaccine uptake in Black communities - Comparison of percentage of Black people at mass vs. remote vaccination clinics	−24,808 at a mass vaccination clinic −1542 at a remote vaccination clinic	- At a mass clinic where individuals were vaccinated with a first or single dose, 3.7% were Black compared to 44% at the remote clinic	- Multi-tiered community approach: engaging faith leaders with the academic community to disseminate information, culturally representative healthcare professional deliver educational webinars with low-barrier access sites to target Black communities to increase vaccination
2	[67]	USA	Black Americans (young adults)	COVID-19	Quantitative, survey	348	- Increased willingness: trust in vaccine information, perceived social approval, perception that other Black people were getting vaccinated, skepticism, and perceived control of contracting virus - Decreased willingness: mistrust in vaccine development, government, and vaccine itself	- Trust and normative perceptions impact Black Americans’ intention to get the vaccine
3	[68]	USA	AA (older)	Flu	Quantitative, survey	620	- 1 out of 3 AA 65 and older in South LA has never received the flu vaccine; 49% were vaccinated within the last 12 months - More likely to receive flu vaccine if recommended by their physician -Less likely to be vaccinated: depression symptoms, lived alone, and experienced a lower continuity of care, satisfaction with availability, access, and quality of care	- Flu vaccination rates in underserved older AA impacted by culturally acceptable and accessible sources - Depression is less likely to be treated in AA; screening and treatment for depression may enhance vaccination in underserved older AA - To reduce vaccine-related inequities health professionals should target those living alone, who are isolated, and suffer from depression
4	[27]	USA	HIV-positive Black adults	COVID-19	Quantitative, telephone survey	101	- Mistrust was significantly associated with higher vaccine and treatment hesitancy - Those with less than a high school education had a higher general mistrust - The most trusted sources were service providers/health professionals followed by local public health officials/agencies, and local government officials - Least trusted sources were the Federal Government and President, followed by social media	- Reception for PH * messages may be increased through healthcare providers and community-based non-political entities
5	[69]	USA	Black Americans	COVID-19	Quantitative, survey	207	- Not wanting to get vaccinated predictors: weak subjective norms for close social network, mistrust of vaccine, e.g., harm and side effects, living in an area with high socioeconomic vulnerability	- Vaccine confidence is lowered by high levels of mistrust for the vaccine - Attitudes of social networks can be influential in encouraging vaccination - PH * communications should acknowledge historical and current racism and discrimination and be clear and transparent about vaccine safety and efficacy
6	[70]	USA	Black adults (89.6% US born)	HPV	Quantitative, data from national surveys 2013–2017	5246	- HPV vaccination initiation was ~1.5× higher in US-born compared to foreign-born - Vaccination was associated with being single in men, some college experience, fair/poor health, obstetric/gynecological visit, and pap test; findings suggest health insurance remains crucial for HPV vaccination	- Vaccination rating for Black immigrants may improve with health insurance. - Promotion should be culturally relevant, age-appropriate, and gender-specific for Black immigrants - To improve prevention measures, HCPs * could highlight the need for eligible males to get vaccinated among foreign-born Black populations
7	[71]	USA	Black Americans	COVID-19	Quantitative, survey	2480	- Police violence concerns were associated with COVID-19 vaccine concerns and worse mental health - Unvaccinated individuals were higher in cultural mistrust, but lower in perceived discrimination, which partially mediated the relationship between COVID-19 race-related concerns and mental health symptoms	- Culturally responsive strategies are needed; individuals may not have the personal agency over factors, such as racism and everyday discrimination, that pose structural barriers to accessing, searching for, and receiving equitable health services
8	[72]	USA	Black Americans that had not received the COVID-19 vaccine	COVID-19	Quantitative, experimental intervention using different messaging strategies. Post-test survey.	739 Black (N = 244), Hispanic (N = 170), white (N = 329)	- Lower VH * was associated with messaging that acknowledged past unethical treat in the medical research of Black Americans and emphasized current measures to prevent medical mistreatment; this was not observed in messaging about the vaccine’s general safety or roles in reducing racial inequities	- PH * vaccine messaging should be tailored to specific concerns and demographic groups
9	[73]	USA	Young Black Americans (18–30 years)	COVID-19	Quantitative, online survey	312	- Those who had vaccine discussions with their family had a more positive outcome expectancy and favorable injunctive norms	-Promoting positive conversations between young Black people and their families could help increase positive vaccination beliefs and decision-making; families could also be a source of misinformation - Increased knowledge about family communications and health among young Black adults could aid in the development of family and network-based interventions.
10	[74]	USA	African American (AA)	COVID-19	Quantitative, preliminary survey, intervention of one of three pro-vaccine messages	394	- Self-persuasive narrative had a more positive vaccine belief with higher vaccination intention - Narrative message and the self-persuasion narrative both had the greatest vaccination intention	- Mass media campaigns should include stories about people who changed their minds about the vaccine
11	[75]	USA	SLE patients (78.4% Black)	COVID-19	Quantitative abstract survey via mail, Internet, and phone	598	- Those VH * were younger, more often Black, less likely to be married, lower income, depressed, less educated, had more Medicare and/or Medicaid, had less trust in the government, doctors, news, lupus advocacy, and support groups, had less general concern for COVID-19, believed in more potential lupus flare ups, lupus-related side effects, and decreased efficacy in lupus - Had received fewer previous flu vaccinations, higher depression, and lower resilience - 66.1% with COVID-19 VH * had a recent flu vaccine	- Flu vaccinations in VH * group show a potential for vaccine receptivity - Outreach led by community leaders and peers should focus on young, Black, depressed individuals with low socioeconomic status
12	[76]	USA	AA	COVID-19	Quantitative, survey	257	- The odds of being vaccine-resistant were 21× higher in participants aged 18 to 29 compared to 50 and older adults; 7× higher in those with housing insecurity, are less likely to be men, more likely to be employed full-time, less likely to have health insurance, lower total number of comorbidities, more likely to be tobacco smokers, less likely to ever have received a flu shot	- Heath systems and organizations must build rapport and trust in vulnerable populations to reduce disparities in vaccine uptake, related infection, and death, through being transparent about COVID-19 vaccines, community engagement, and diversity in medical professionals
13	[77]	USA	Black/AA	Childhood immunizations	Quantitative review, data from UW health systems, Google and Google Trend searches; 2015–2020	University of Wisconsin, Madison serves over 600,000 patients each year	- Child immunization rates in BAA communities are continuously declining - Media review suggests anti-vaccination leaders have increasingly targeted the BAA community with misinformation and skepticism - Main questions BAA parents have including safety and information, such as pros/cons, about vaccines	- Health systems must be assessed for disparities and drivers to effect change; health systems and professionals must look to understand and address the fears increased by anti-vax leaders - Strategies should combat negative media campaigns (anti-vax) and close knowledge gaps - Healthcare organization must fund their communities and public health departments to build trust and decrease disparities
14	[10]	USA	AA with heart failure (HF)	Influenza (flu)	Quantitative, survey, during 18 February 2017 flu season	152	- Predictors of vaccination: 55 and older, increased number of comorbidities, received flu vaccine information and recommendation from HCP *, especially their cardiologists - More patients received flu vaccination information and recommendation from their internists and family physicians than their cardiologists; maybe due to frequency of visits, but all had consulted with their cardiologist at least once	- Physicians have a crucial role in positively influencing their patients’ vaccination behavior; they must be aware of this influence and consistently provide flu vaccine education and recommendation during consultations and outpatient visits - It is recommended that there be standing orders and protocols in the electronic medical record system to allow HCPs *, such as nurses, to recommend the flu vaccine and to vaccinate patients without a direct order from a physician or supervision when an assessment for a true medical contraindication is not needed - Flu vaccination concerns should guide the tailored education of patients, which may improve coverage rate in patients with high-risk conditions
15	[78]	USA	AA hospitalized with severe COVID-19 infection	COVID-19	Quantitative abstract, phone survey and data from medical records	48	- 66% of the patients would not get the vaccine when available, despite comorbidities. - Reasons for declining: fear of vaccine side effects (61%), distrust of the pharmaceutical companies that make vaccines (58%), and uncertainty about effectiveness (42%) - 75% of the participants were more likely to accept it if their primary care physicians or specialists recommended it; 8% would accept it based on information in TV/radio ads or the Internet	- Education that is focused on patient concerns and direct recommendations from medical providers may increase vaccination coverage in vulnerable populations
16	[79]	USA	AA hospitalized with COVID-19 infection	COVID-19	Quantitative, survey post recovery and discharge	119	- Higher likelihood of acceptance: male and uninsured - Higher likelihood of declining: patients with congestive heart failure, coronary, artery disease, diabetes mellitus, and hypertension - Major reasons for declining: combination of distrust in efficacy despite research findings and distrust in pharmaceutical companies that produce vaccines (78%), fear of side effects (65%), perceived immunity against re-infection (29%). - 3/10 AA patients who recovered from infection would accept a “safe and effective” COVID-19 vaccine	- Medical providers and community-based advocacy groups should work together to build trust and dispel misconceptions
17	[80]	USA	Black churchgoers	COVID-19	Quantitative, pretest survey, intervention with a 1.5h webinar, post-test survey	220	- Most of participants personally knew someone who became infected; few were concerned about hospitalization if they became infected. - Many participants: learning facts about COVID-19 was most impactful and hearing from Black physician researchers who were involved in the vaccine’s development -Webinar increased willingness -Willingness was higher in Black males, no age difference -VH * may be reduced in high-risk groups through community academic partnership collaborations to reach Black communities. -May influence likelihood: changes in perceived benefits, susceptibility, and seriousness	- Excellent virtual tools to reach large audiences; during pandemic social interactions restricted - Intervention initiatives could be strengthened through longstanding relationships with the community.
18	[81]	USA	African American parents	HPV	Quantitative manuscript. pretest, randomly assigned a pamphlet with arguments for vaccination with either a gain-framed or loss-framed message, followed by a post-test	184	- Loss-framed messaging with parents that had a low perception of the HPV vaccine efficacy meant that they were more reluctant to vaccinate their children than those who viewed gain-framed messaging - Defense-motivated process and psychological reluctance occurs with loss-framed messaging, which was it was less persuasive than gain-framed messaging	- To prevent reactance and message rejection to tailored messages, people’s perceptions of vaccine efficacy should be assessed before exposure - The relationships between individual perceptions, message framing, and reactance in the context of HPV vaccination is important to understand, particularly among AA, who are disproportionately affected
19	[82]	USA	AA	COVID-19	Quantitative, survey	428	- 48% of unvaccinated participants reported being VH *; younger, from northeastern US, Republican political affiliation, and religion other than Christianity or atheist	- Large amount of variance in the likelihood to get vaccinated for COVID-19 in VH * AA
20	[83]	USA	Older adults (AA 80.5%)	Pneumonia	Quantitative, pre-test survey, intervention Pharmacists’ Pneumonia Prevention Program (PPPP); 4 domains (1) pharmacists and pharmacies, (2) vaccination, (3) pneumococcal disease, and (4) physicians, followed by post -test survey, 3-month post intervention survey	190	- 21% completely agreed with the statement that the pneumococcal vaccine would prevent pneumonia at baseline; this more than doubled following the program, but returned to baseline after 3 months - 16% trusted pharmacists as immunizers at baseline; nearly half of participants trusted a pharmacist to provide them with a vaccination at post-test, and 27% did 3 months after the program -Trust in pharmacists remained lower than physicians throughout the study	- Individuals have many encounters with healthcare throughout their lives that may influence their beliefs; therefore, altering beliefs may require sustained efforts - Pharmacists and seniors’ centers partnering can be an effective model for community engagement; should consider support for pharmacist’s educational services
21	[84]	USA	Black Americans	COVID-19	Quantitative, survey	1040, matched 2010 US census demographics	- Black Americans had higher levels of VH *; medical trust decreased VH * - Health officials and media risk conflating medical trust, conspiracy thinking, and demographics when seeking to understand perceptions about the COVID-19 vaccine	- Community engagement and dialogue may help promote COVID-19 vaccine acceptance - Structural racism likely the cause of racial immunization disparities
22	[85]	USA	Black adults	COVID-19	Quantitative, phone survey	350	- 48.9% of Black adults in Arkansas were not COVID-19 vaccine hesitant - 22.4% were very hesitant, 14% somewhat, and 14.7% a little hesitant; hesitancy was 1.70× greater for Black adults who experienced a COVID-19 related death of a close friend/family member, 2.61× greater in those who reported discrimination with police or in the courts, and hesitancy was negatively associated with age	- Among Black adults, there may be a link between COVID-19 VH * and racial discrimination in the criminal justice system - Further research is needed to determine if there is a causal relationship between death caused by COVID-19 and VH *
23	[86]	USA	Black Americans	COVID-19	Quantitative, survey online or by phone, measured at first and second wave	889	- Trust in information sources is associated with vaccination beliefs - Differences in trust do not account for the differences in vaccination beliefs by race - Race influenced the relationships between trust in Trump and PH * officials and agencies and vaccination beliefs; effects of trusting these sources on COVID-19 vaccine-related beliefs are less among Black participants; trust in these sources is less consequential to their pro-vaccination belief. - Some sources are more likely to be trusted by Black Americans, such as social media; trust in certain sources is associated with lower VH *	- Beliefs can mediate the association between vaccination intention and race, moving the focus from intention to beliefs (VH *) - The observed relationship between vaccination beliefs and race is not explained by trust in information sources alone - Trusted sources could be used to mitigate the effects of the misinformation and to communicate pro-vaccine information
24	[87]	USA	Black Americans who were eligible for but had not received COVID-19	COVID-19	Quantitative, survey	1278	- Black Americans’ vaccination intention was independently and interactively affected by the social norms - The norms of all Americans are a decision basis for Black Americans to make decisions regarding COVID-19 vaccination - Social norms of an ethnic group predicted of vaccination intention and could be a potential group to target for interventions	- Practices of comparing and contrasting racial differences in COVID-19 vaccination rates and interventions using strong social norms for all Americans to mobilize Black Americans should be carried out cautiously - Identifying an influential reference group other than a close social network is meaningful and informative for promoting COVID-19 vaccination among Black Americans - Lower vaccination intention among those with perceived strong social norms for all Americans may be due to perceived herd immunity, giving a false sense of protection due to high vaccination and infection rates in the society while Black Americans who were hesitant from the beginning may be even more resistant due to a perception that the vaccine is unnecessary

* VH = vaccine hesitancy; HCP = healthcare provider; SDOH = social determinants of health; PH = public health.

#### 3.1.3. Mixed Methods Data Extraction

**Table 3 vaccines-12-00269-t003:** Data extraction from mixed methods articles. The first author and date of publication, country, population, vaccine, design sample, size, findings, and conclusion relevant to the research question for each of the evidence sources.

Number	Article	Country	Population	Vaccine	Design	Sample Size	Findings	Conclusion
1	[88]	USA	AA (African Americans) youths	All vaccines	Mixed methods protocol Phase (1) qualitative IDI (in-depth interview) to assess (2) quantitative, based on assessment (3) quantitative assessment of intervention—expected completion 2023	Phase 1 and 2: (N/A). Phase 3: (4 clinics and 120 AA youths)	- An intervention that has the potential for PH * impact emphasizes youth autonomy and decision-making; interventions that leverage existing infrastructures increase the likelihood of their transferability - Tablet-based interventions with tailored messaging for youths, including motivational interviewing, text reminders, and with rural context, can be designed to reduce HCP * burden with consideration to environmental and practice limitations	- Researching and creating effective interventions that target AA youths can provide information about VH * and potentially inform effort by practitioners and providers
2	[89]	UK	African ancestry with HIV	COVID-19	Mixed methods questionnaire with Likert scale and free text, poster	540 from 9 sites	- Unvaccinated participants more concerned about side effects and what is in vaccine, such as microchips and materials from pigs or fetuses - Persuade individuals to vaccinate with informed choice through full discussions on trial data and full disclosure of results, 100% efficacy against COVID-19, more data on long-term effects, such as fertility, choice of vaccine, mandatory, such as for travel, and a single dose - Concerns include bioweapon technology, irreversibility altering DNA, medical history, and religious concerns based on what is in the vaccine	- High COVID-19 vaccine uptake was found; those not vaccinated had high levels of concerns and low vaccination necessity - Community engagement can help address health inequities, vaccine concerns, and misinformation
3	[90]	USA	AA undergoing smoking cessation treatment	COVID-19	Mixed methods, secondary analysis of data from ongoing RCT (random control trial), questionnaire with closed and open questions	172	- Few participants mentions a physician’s opinion as being influential to their decision get vaccinated for COVID-19; most mentioned information that would be helpful in decisions: efficacy, safety, side effects, initial outcomes of others - Participants with low vaccine intentions had concerns about vaccine development, trustworthiness, and efficacy	- PC and the medical system need a concerted effort to gain trust in AA communities; carry a high burden of COVID-19
4	[91]	USA	AA smoking cessation	COVID-19	Mixed method, survey with open and closed questions, baseline and follow-up, poster abstract	172	- 36% not willing to take the COVID-19 vaccine if freely available, most common reasons are a lack of trust in the vaccine, uncertainty due to it being rushed, unsure what was in it, not enough studies, and not taking vaccines in general, including flu shot - Willingness to get vaccinated for COVID-19, most common reasons were protection, to feel safe, as well as worry of getting sick or dying from infection	- High rates of HV could prolong higher negative impacts from COVID-19 on AA
5	[92]	USA	Black Americans	COVID-19	Mixed methods, survey, interviews, focus group discussions (FGD)	183 surveys, 30 interviews, 8 FGD (n = 49)	- Potential factors affecting access perceived to be transportation barriers; no Internet access for relevant COVID-19 information or to register for appointments; Internet could be difficult for some elderly people to navigate; healthcare system is complex and can be difficult to navigate - Factors affecting vaccine acceptance were perceived as communal safety, peer pressure, fear fatigue of worrying for oneself and others, mandated pressures, individual’s perceived risk of infection and severity of outcome, belief in other effective means of protection; vaccine confidence in safety and efficacy associated with increased willingness to get vaccinated, those familiar with vaccine immunology were hopeful that more information would increase others’ willingness; more willing to get vaccinated once with observed effectiveness in others - Factors associated with vaccine resistance: deep-rooted belief, e.g., religious, conspiracies theories and myths, distrust in the government, and level of trustworthiness of healthcare system and medical community - Improve vaccination rates: community outreach and navigators, vaccine endorsement (scientists, community leaders, clinicians, peers), timely information in multiple formats, community vaccination cites (churches, community centers), testimonials by trusted professional, HCPs * from community as frontline advocates	- VH * changes with situational context and knowledge - Could reduce accessibility issues with community collaborations to establish community-based clinics and by employing community navigators and coordinators
6	[93]	USA	AA	COVID-19	Mixed methods, questionnaire with open and closed questions	203	- VH * mainly due to mistrust in the healthcare system, other reasons: the speed of vaccine developments, confidence in current health and lack of information, mistrust in the healthcare system and government - Non-hesitance was mainly due to already being vaccinated, the protection of self and others, required by school or work - Some stated that encouragements from trusted individuals may change their minds about being vaccinated	- HCP * and pharmacists can contribute to improving confidence and decreasing vaccine hesitancy
7	[94]	USA	Black Americans who expressed VH	COVID-19	Mixed methods, IDI, qualitative thematic analysis and quantitative code application	18	- All stated lack of trust in the government regarding information dissemination and doubt in approach to healthcare - Most were concerned that the vaccine was rushed, stated more time and data were the most important intervention to increase their willingness to be vaccinated; other factors included seeing friends and family vaccinated, but celebrities and politicians would not sway them; other factors included being able to go out to events with friends and family, personalized medical advice from a physician, more information about people who were vaccinated, and emphasis on protecting friends and family - Many had a distrust for vaccines in general, fear of long- and short-term side effects, and question those distributing vaccines to Black communities caring about them based on historical medical mistreatment - Physicians were the most frequent sources of information; other sources were newspapers, television, church, virtual town halls, hospital websites, family and friends, and a few used social media	- 1/3 of those that expressed VH * had become vaccinated by the end of the study, potentially indicating fluidity in opinions regarding VH * - Prioritizing vaccine acceptance in communities most susceptible to infections helps protect the larger population and, thus, individuals nationwide

* VH = vaccine hesitancy; HCP = healthcare provider; SDOH = social determinants of health; PH = public health.

#### 3.1.4. Commentary Data Extraction

**Table 4 vaccines-12-00269-t004:** Data extraction of commentary articles. The first author and date of publication, country, population, vaccine, design sample, size, findings, and conclusion relevant to the research question for each of the evidence sources.

Number	Article	Country	Population	Vaccine	Design	Sample Size	Findings/Key Statements	Conclusion
1	[95]	USA	Black women	COVID-19 and influenza (Flu)	Commentary	N/A	- The healthcare industry must directly engage to foster authentic relationships and rebuild trust as the potentially lethal combination of COVID-19 and influenza create the possibility of a “twindemic” - Consistently low flu vaccinations in Black women foreshadows concerns with COVID-19 vaccination; top flu vaccine concerns were adverse effects, safety, and effectiveness - Low vaccine uptake in Black women not unexpected based on historical mistrust in healthcare system from Tuskegee syphilis experiments to underrepresentation of Black people in vaccine trials - Healthcare should be view as preventative of negative health impacts of the SDOH *	- Black people will not have equitable access to cures without recruiting more Black people in research and vaccine trails - Mistakes of the past are doomed to repeat without community empowerment
2	[96]	USA	Black people	COVID-19	Commentary	N/A	- Everyday racism, such as denying pain, treatments withheld, and misdiagnosis, are ignored when citing historical mistrust at the cause for mistrust in the HC system - Trust is critical; Black women prefer Black physicians, even waiting months for appointments - Concordant health messaging is important	- Vaccine rollout needs Black physicians and investigators at the forefront - Black health leaders need to give public health messages - Black scientists relate to the needs of communities
3	[97]	Canada	Black people	COVID-19	Commentary	N/A	- Politicization of the vaccine, lack of culturally relevant information and accurate and timely race-based data, poor public health coordination response between levels of government added barriers and contributed to mistrust - To address VH *, engage Black community leaders in all steps of vaccine development, distribution, and monitoring to improve transparency through an internal assets-based approach - Increase trust and transparency due to vaccine’s quick development including risks of side effects, efficacy, such as for children and pregnant women, address misinformation and conspiracy theorists that exploit historical injustices, ensure diverse representation in vaccine trials - Ensure clinics in rural communities since travel can be timely and costly - Long-term support to address needs and concerns after vaccination	- Strategies must address concerns and fears of racism in vaccine development and distribution
4	[98]	USA	Black people	COVID-19	Commentary	N/A	- The difference in experiences of Black and white Americans with the healthcare system may account for differences in trust in the government and medical establishment involved in vaccine development - Reasons to fear and mistrust based on medical history - Be transparent to combat mistrust, comprehensive communication, acknowledge uncertainty, increase accessibility, ensure vaccination is not cost-prohibitive - PH * officials and medical professionals should communicate the safety and efficacy of vaccines due to high levels of suspicion of pharmaceutical companies—transparency about delays and side effects would help build trust	- Black Americans have had worse health outcomes than white Americans across many conditions for decades—COVID-19 can be used to eliminate disparities, restore trust by listening to most disadvantaged, acknowledging reasons for mistrust, and maintaining transparency for racial justice
5	[99]	USA	Black Americans	COVID-19	Commentary	N/A	- A Black intensive care nurse was the first American to receive the vaccine with the hope of inspiring others - Increased uptake attributed to outreach by health leaders, medics, and faith and community organizations, such as through livestream town halls; community support, peer influence, and testimonials make a difference. - Increased access and availability with more locations within the community, like barbershops and hair salons, especially helps those of lower socioeconomic levels to minimize travel - Medical professionals should remind patients of the importance of vaccines - Mistrust drives VH *, those who were against are starting to consider it, misinformation that targeted Black communities caused a plateau, but increased when people saw pain and suffering, and vaccines were mandatory—policies should require all populations to be vaccinated for PH *	N/A
6	[100]	USA	Black Americans	COVID-19	Commentary	N/A	- Acknowledge historical mistreatment by medical establishments, the trauma it caused, and health disparities from unconscious bias and racism, and do not blame Black Americans themselves - Mutual and transparent conversations between physicians and Black patients to listen to concerns and acknowledge VH *	- Create health equity and anti-racism initiatives, including misinformation, availability, accessibility, roundtables to bring together institutions and community leaders, educational committee to develop education with consistent and accessible messaging to target vulnerable populations, provide transportation, assisting with neighborhood and mobile vaccination distribution
7	[2]	Canada	Black people	COVID-19	Commentary	N/A	- VH * due to misinformation of literacy gaps and medical distrust and structural racism - Afrocentric health promotion and counselling improved uptake of flu and COVID-19 vaccines and center on clients’ values and perspective - Clinicians can bridge gaps and improve vaccine uptake with communication framework - Outreach and confronting anti-Black racism to increase vaccine confidence can be improved with Black-led partnerships between trusted stakeholders and healthcare - Clinicians should support patients to navigate complex systems, state data on the number of vaccinated Black clients and Black scientist’s contributions, emphasize the importance of population-wide coverage, and offer accurate, current information to high-risk Black patients about access - Provided greater access and/or be on priority lists	
8	[101]	USA	AA (African Americans)	Flu and COVID-19	Review		- Low flu vaccine uptake is at least partly from bias in medicine causing mistrust, safety concerns, and barriers to access, exacerbating adverse health outcomes - Policy initiative and robust education to build trust in the health benefits of the flu vaccine and ultimately to build trust in the COVID-19 vaccine when it becomes available - Increased vaccine trust and confidence, including perceived health benefits and recommending it to others, is associated with increasing racial fairness - Must provide evidence-based information on the COVID-19 vaccine to support acceptance once available - Mobile vaccination programs can rapidly disseminate and increase access in hard-to-reach communities; unified service providers can increase mobile infrastructure	- To increase COVID-19 vaccine uptake, establish new initiatives to build trust and support evidence-based medicine - To support health equity, address racial/ethnic and socioeconomic disparities -AA voices must identify potential barriers in Black communities and overcome historical mistrust in clinical trials
9	[102]	USA	AA	COVID-19	Review	N/A	- Increased risk for severe illness from COVID-19 due to comorbidities, and SDOH *, such as lower health literacy and educational attainment, essential workers, inadequate healthcare access and utilization, living in crowded housing, and lower income reduces the likelihood of following methods for mitigating risk recommended by health officials - Parents are buffers/filters for their children due to social dispositions, and racial socialization related to racial identity and systemic racism and medical abuse is well known; mistrust is racially socialized from integrated experiences and trauma, such as the impact of others being mistreated in healthcare; child witnessing parent may prevent them from seeking care - Peer relationships (including family) can increase susceptibility to conspiracy theories and have been associated with increased VH *, such as that the vaccine has malicious intent against people of color - Have expressed hesitancy about the vaccine safety, effectiveness, and speed of development - Online misinformation target AA communities; the source cannot always be determined; others may masquerade as AA - Vaccine uptake is influenced by events, such as the Black Lives Matter movement, and the impact of the pandemic, such as disruptions to work and school, which increased anxieties	- Service providers need to act as partners in assessing, advocating, and implementing interventions for disparities while addressing the SDOH *
10	[103]	USA	Black community	COVID-19	Commentary	N/A	- Have the highest rate of individuals that are unsure or will never get vaccinated; concern that VH * may further heath disparities - Distrust in government and medical professionals is justified by historical racism in medical research - Systemic and structural barriers include distribution, access, availability, and transportation - Black men are underrepresented in the medical profession, and they are critical for building trust; build trust by increasing the number of Black nurses, physician, dentists, pharmacists, and allied professionals - Access to health information is key to educating and dispelling false information on social media that can influence decision-making and hesitancy; build trust by providing information with clear, lay language to understand health related information and answer questions through online and physically locations, such as barbershops, churches, mosques, and other trusted community-based organizations	- The Black community should be vaccinated, and medical professionals must do more to create trust and improve care
11	[104]	USA	Black Physicians	COVID-19	Commentary	N/A	- Racial inequities in vaccine rollout echo the disparities experienced throughout the pandemic with less vaccines going to Black residents - Mistrust has been brought up as a factor against low vaccine uptake in Black communities; a commonly proposed solution is positioning Black physicians and investigators at the forefront to provide racially concordant messaging and build trust; this circumvents rather than deals with the issues of mistrust - Black physicians have a greater burden then their non-Black counterparts as they earnestly work to encourage vaccination in Black communities; they themselves are underrepresented in medicine because of racism - First-hand experiences and collective experiences of family, community, and history are not simply reflected in the physician– Black patient relationship but in the institution’s relationship to the Black community; the institution cannot substitute an individual Black physician’s trustworthiness for its own	- Placing Black physicians as the solution deflect the responsibility of institutions and generates systemic problems for those already overburdened; solutions to present racial inequities must not exacerbate the problem in the future
12	[105]	USA	Black people	COVID-19	Commentary	N/A	- A physician, researcher, epidemiologist, and Black woman had COVID-19 when it first came out, and they reviewed the immunology, listened to interviews, and reviewed the data before getting vaccinated, still with uncertainty; experiences of structural racism overpowered medical education - If systemic racism cannot be understood, it will be difficult to understand VH * in the Black community - Service providers need to acknowledge wrongs and failure to gain trust based on historical and everyday racism, and undergo training to do better	- The government and medical community should work towards quick and equitable distribution to build trust
13	[106]	USA	Black patients and physicians	COVID-19	Commentary	N/A	- Fewer providers that look like them due to the healthcare infrastructure may lower intention due to general lack of trust - Being a Black physician does not remove concerns of mistreatment and distrust in the system; VH * still exists - Fears driven from historical medical experiments and stories passed down through generations are the basis for distrust in the healthcare system - When healthcare leaders recommended mass vaccination to Black communities, could cause fears of further being tested on before the general population - Focus should be educating the population rather, not pressuring an individual - Community and healthcare leaders need to leverage tools, like social media. to offer advice - Every physician must counsel each patient and address concerns about vaccination - Questions about the novelty of the technologies in the COVID-19 vaccine must be answered, with full disclosure to earn trust	- Properly convey truth that the vaccine will not be tested on vulnerable populations and is administered safely for their benefit; this may be emphasized by witnessing the safe administration of the vaccine on societal leaders, such as healthcare professionals and politicians - Vaccine transparency is a short-term solution, more healthcare workers are needed that represent their best interests, understand their experiences, are from their neighborhoods, and are as diverse as they are
14	[107]	USA	Black people	COVID-19	Commentary	N/A	- Historical mistreatment and current concern causes distrust of vaccines and institutions - Trusted messengers are crucial to protecting the community, e.g., pastors - A proposed framework to build trust and acceptance includes understanding history and context, creating partnerships with shared responsibility and power, listening and empathy, engaging pastors as trusted messengers, and co-creation of solutions with faith leaders and their community, governments, and institutions to create sustainable, long-term change	- Pastors and others in the faith community must work with governments and institutions to build trust, inform, facilitate discussions, and create measurable improvements - Sustainable efforts for lasting and impactful change are needed to establish, stronger collaboration, build relationships - Possible vaccine solutions may be initiated by focusing on issues important to the community
15	[108]	USA	AA hemodialysis patients	COVID-19 Influenza (flu)	letter to the editor	90 (83% AA)	- Half of the hemodialysis patient survey participants were willing to get the COVID-19 vaccinations; influenza vaccination was associated with willingness - Previous personal history of COVID-19 was not associated with accepting the vaccine, low willingness to receive the vaccine may pre-exist the pandemic based on interactions with HCP *, low access, VH *, low clinical trial participation, and cost - HCP * must collaborate with trusted partners to build trust, identify the best alternatives to improve quality of care, and to help create targeting messaging around COVID-19 - VH * increased by not enough safety and efficacy information	- Patient education is essential to increase vaccine uptake and combat misinformation
16	[109]	USA	Black Americans	COVID-19	Commentary	N/A	- Black HCPs * role in providing care for their patients and their communities - Exploitation and persecution of Black Americans by the US healthcare system has affected Black communities for generations; COVID-19 gives an opportunity for the healthcare system to begin to make amends for historical injustices and discrimination - Strategies to address VH *: acknowledge past and present injustices based in policies, all clinicians and trainees have iterative cultural competency training with a focus on the SDOH * for health equity, transparency in risks and benefits and accountability of vaccine delivery, developing messages that are educational, informative and acknowledge as well as addressed apprehension about safety and side effects in a culturally sensitive way, partner with trusted sources, such as faith-based organizations, political advocacy groups, and grassroots organizations to engage Black communities in a personal and culturally sensitive way, increase access by reaching communities when and where they prefer to be vaccinated	- Address VH * by building an equitable mechanism through genuine communication, thoughtful partnerships, relevant messaging, and removing barriers
17	[110]	USA	African Americans and others in marginalized communities	COVID-19	Commentary	N/A	- Differences in the rate of vaccine uptake may be explained by external factors, such as structural racism - Healthcare systems must mitigate factors that prevent making informed decisions about COVID-19 vaccination; scientists, researchers, and physicians must understand factors outside of the person that explain decisions and behaviors related to vaccination - Historical basis for mistrust and cynicism towards health professionals due to historical events, such as the Tuskegee experiment, cannot be overlooked by healthcare policy makers - Being Black should not be a predictor for VH *; further analysis into the ways that social and structural determinants explain COVID-19 vaccination related choices is needed	- Methods of addressing health disparities that existed before, and were exacerbated by the pandemic include repairing mistrust, increasing AA healthcare workers, including in the community, providing training and incorporating culture-based healing practices into services and research, developing and assessing the cultural competence of healthcare workers, mobilizing community workers to provide COVID-19 information about safety and efficacy to gatherings, such as in churches and recreation areas, community engagement research, and ensuring protection from unethical research practices including informed consent
18	[111]	USA	Black people	COVID-19	Commentary	N/A	- Distrust is deeply rooted in centuries of racist exploitation by American physicians and researchers, not only in Tuskegee - The main responsibility of overcoming racism is not on Black people themselves - Pharmaceutical companies can build trust by ensuring that that a vaccine is not submitted for approval until it has been thoroughly assessed for efficacy and safety, especially due to the politicization of vaccine trials, and that trial participants will receive medical care if injured as a result of an experimental vaccine, as well as informed consent, including all aspects of the trial, to maximize transparency; participants have the right to expect that Black communities will have fair access to results	- The COVID-19 vaccine’s success will depend on trust that it is safe and effective and the trustworthiness of organizations offering them. - Efforts must have bidirectional collaborations with communication and learning grounded in the grassroots involvement of organizations and individuals trusted by Black community members and leaders
19	[112]	USA	Black people	COVID-19	Commentary	N/A	- Asking why Black people do not trust HCPs * pathologizes them as having something wrong rather than the conditions around them; it ignores the roles health institutions have in distrust, and implies that many of their health-related problems would go away if they were more trusting - Women are often primarily tasked with care work; those who are the most vulnerable, such as Black women, care for others, often with low wages and little protection, while simultaneously needing care - Asking questions about vaccine safety and efficacy is prudent and diligent, rather than hesitant	- Reluctance is justifiable and using an explanatory narrative may overvalue the role of trust; it should be a part, but not all of, the conversation.

* VH = vaccine hesitancy; HCP = healthcare provider; SDOH = social determinants of health; PH = public health.

### 3.2. Analyzing the Evidence

Thematic mapping [49] (Figure 2) was used to collate and summarize the extracted data, as it is ideal to accommodate the multiple types of evidence sources in this study. This method also allows for the production of themes while maintaining the breadth of information within each theme through the creation of subthemes (descriptive and analytical themes), which is consistent with [48], which has stated that a ScR seek to explore the breadth of existing evidence sources and not to qualify the evidence or provide generalizable robust findings and Peters et al. [32] which has stated that results should be descriptively mapped rather than synthesized. The initial step of quality appraisal was not performed as it is not necessary for a ScR [32]; therefore, the evidence sources were divided out my type during data extraction. 

#### 3.2.1. Thematic Mapping Diagram

The three phases of TM are (1) individual analysis, (2) within-group analysis, and (3) across-group analysis.

**Figure 2 vaccines-12-00269-f002:**
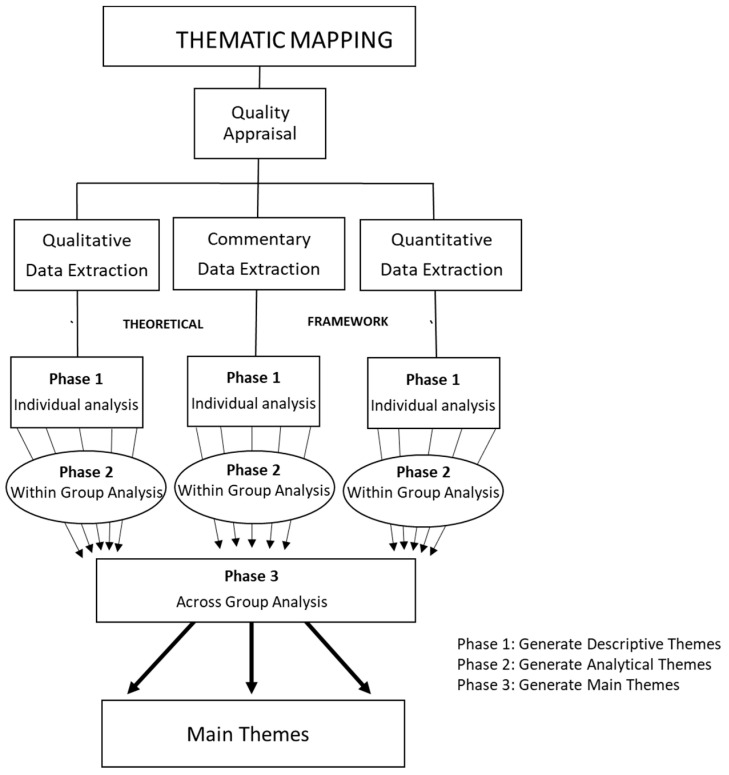
Thematic mapping. This diagram from Etowa et al. [49] illustrates the steps in thematic mapping. These steps were followed in our study except for the qualitative appraisal which is not required in a scoping review [32].

#### 3.2.2. Phase (1): Individual Study Analysis

The aim of Phase 1 was to create initial codes and descriptive themes which are informed by the theoretical underpinnings used in this study, namely, intersectionality, the social determinants of health (SDOH), and the socio-economic model (SEM) as they relate to low vaccine uptake in African, Caribbean, and Black (ACB) populations relative to public health in high-income countries. In this phase, the articles were divided by article type, which was found to be qualitative, quantitative, mixed methods and commentaries. To ensure methodological rigor, two reviewers independently coded the extracted data from each article; MA and SB extracted the qualitative articles, GO and SB extracted the quantitative articles, KS and SB extracted the commentaries, and RS and SB extracted the mixed methods articles. Consistent with Objective 1 of this study, which is to identify concepts and boundaries of existing evidence sources on low vaccine uptake in ACB populations, the generated codes from each type of article was used to create descriptive themes which then further defined the context and boundaries of concepts within each group. The descriptive themes were generated based on grouping the codes by similarities and differences until a consensus was reached. JE worked with the entire group through a participatory data analysis process to create the descriptive themes.

#### 3.2.3. Phase (2): Within-Group Analysis

Within Phase 2, analytical themes were created to gain further insight into the characteristics of the descriptive themes that reflected the content within each type of article grouping (qualitative, quantitative, mixed method and commentaries) separately. The descriptive themes from each article were grouped based on similarities and differences into analytical themes through induction and consistent interpretation with the PCC research question and the theoretical underpinnings as in Phase 1. The iterative process of creating the final analytical themes concluded when a consensus was reached between the research team members. 

#### 3.2.4. Phase (3): Across-Group Analysis

Within Phase 2, main themes were created to give a broad overview of findings based on the similarities and differences across all groups (article types). 

These main key themes were used to map the evidence on concepts and boundaries, identify research gaps, and determine existing strategies to increase low vaccine uptake in the study population (Objective 3). The themes developed are recorded in the thematic map (Figure 3).

### 3.3. Presentation of the Evidence 

Thematic Mapping results (See Appendix A for larger diagram). 

The four main themes of (1) inequities and racism, (2) sentiments and behaviors, (3) knowledge and communication, and (4) engagement and influence were developed by using thematic mapping for analysis to identify the concepts and boundaries of existing evidence sources on the determinants and interventions of low vaccine uptake in ACB populations relative to healthcare in high-income countries; through these results, gaps in the research will also be identified.

**Figure 3 vaccines-12-00269-f003:**
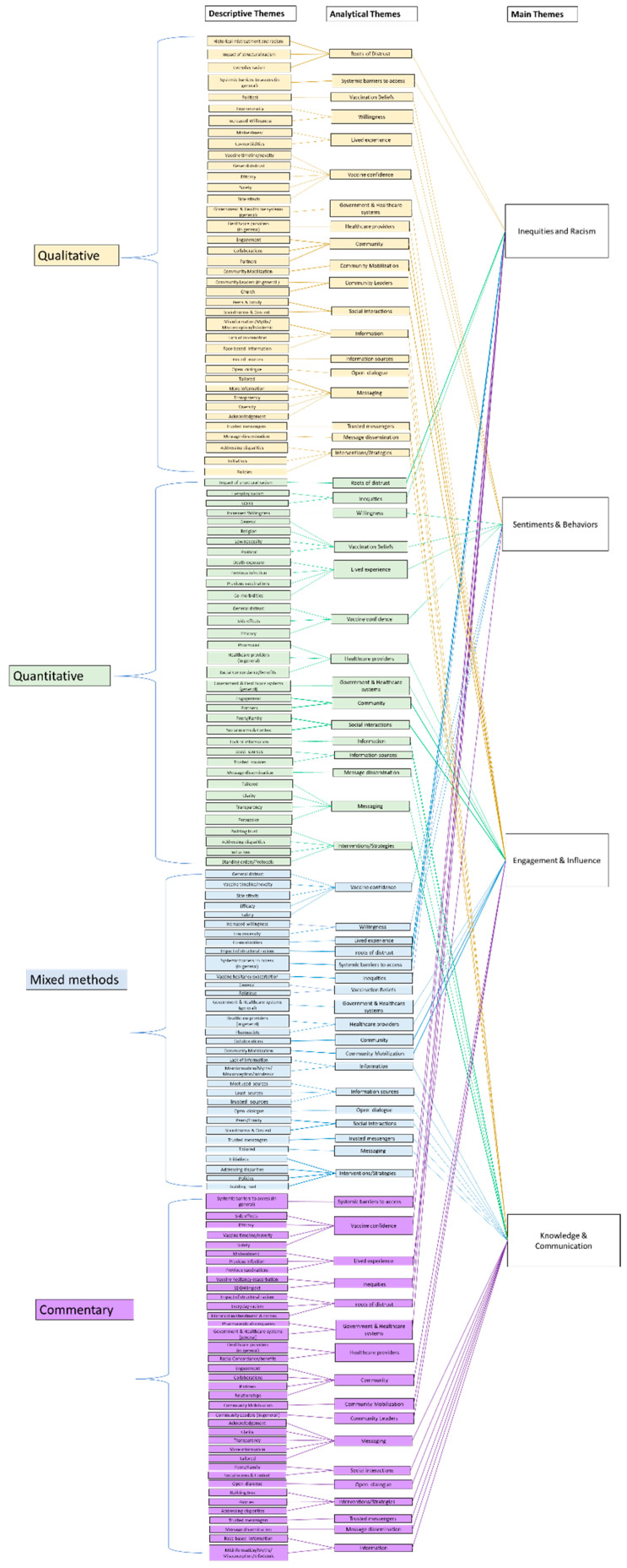
Results of the thematic mapping. This shows the range of subthemes within each main theme as well as the composition of different types (qualitative, quantitative, mixed methods, and commentaries) of articles relating to each theme and subtheme, to help define concepts and their boundaries.

### 3.4. Theme 1: Inequities and Racism

The main theme of inequities and racism refers to a lack of fairness due to discrimination, based on race. This main theme is further subdivided by the analytical themes of inequities, roots of distrust, racial burden, and systemic barriers to access, which further expand on the breadth of this overarching theme.

#### 3.4.1. Inequities

The analytical theme of inequities in this ScR refers to a lack of fairness in association with the descriptive themes of vaccine hesitancy (VH), the social determinants of health (SDOH), and SDOH impact.

#### 3.4.2. Vaccine Hesitancy Exacerbation

With higher rates of individuals indicating that they are either unsure or unwilling to get vaccinated, there is concern that “vaccine hesitancy” (VH) may further perpetuate the health disparities already experienced within Black communities [91,103]. In addition, these concerns were further heightened during flu season, when suboptimal flu vaccine uptake could exacerbate the adverse health outcomes similar to the effects of COVID-19 [101]. 

#### 3.4.3. Social Determinants of Health

The demographics of those expressing VH are reflective of the social determinants of health (SDOH) with higher rates of housing insecurity [76], lower income [75], full-time employment [76], health insurance status [76], less education [75], living in areas with high socioeconomic vulnerabilities [69], more likely to smoke tobacco [76], not married [75] or living alone [68] fewer comorbidities [76], to have depression [68,75] lower levels of resilience [75] less than a high school education [27] and be influenced by age, whereby those 18-26 were reported as being less likely to initiate the HPV vaccine compared to those younger [70] and those 18-29 were 21× higher in being vaccine resistant towards the COVD-19 vaccine compared to those 50, and older respectively [76]; Willis et al. [85], and Sharma et al. [82], also found a negative association between COVID-19 vaccine hesitancy and age. In addition, Black vaccination initiation (HPV) in the US was lower in those born in other countries, with immigrants from the Americas/Caribbean Islands having a lower rate than those from Africa [70] and Sharma et al. [82], reported that VH rates differed by geographic regions. 

Those with higher vaccine acceptance had come college education [70], were in fair or poor health [70], uninsured [10], increased comorbidities [10], male (COVID-19, Flu) [10,80] and single male (HPV) [70], 55 and older (Flu) [10]. HPV vaccination was associated with obstetric/gynecological visits and pap tests [70].

#### 3.4.4. Social Determinant of Health Impact

Being ACB should not be a predictor for VH; social and structural determinants must be further explored so that underlying issues are addressed [110]. ACB people are at elevated risks for severe COVID-19 related illness from the SODH including being essential workers, having lower health literacy and educational attainment, having inadequate healthcare access and utilization, and living with low incomes and crowded housing, which makes following risk mitigations recommended by health officials more unlikely [102]. Furthermore, health outcomes have been significantly worse than their White peers across various conditions for decades [98]. 

#### 3.4.5. Roots of Distrust

The roots of distrust refer to the reason stated in the literature for the distrust felt within ACB populations. This analytical theme included the descriptive themes of historical mistreatment and racism, the impact of structural racism, and everyday racism.

#### 3.4.6. Historical Mistreatment and Racism

Historical mistreatment and current concerns have caused ACB communities to be highly distrustful of vaccines and institutions [107]. Distrust in the healthcare system is based on fears driven by historical medical experiments and stories based down through generations [106], and is a primary driver of VH [99]. Furthermore, the exploitation and persecution of Black Americans by the US healthcare system has impacted Black communities for generations [109], where systemic racism and medical abuse are well known in ACB communities [102]. The low vaccination rates of Black women are alarming but not unexpected based on historical mistrust in the healthcare system ranging from the Tuskagee syphilis experiments to the current underrepresentation of Black patients in vaccine trials [95]. Although the root of ACB distrust is often traced back to Tuskegee, racist exploitation by American physicians and researchers has been occurring for centuries [111]. Distrust in government and medical professionals is also well-founded and justified by the historical racism within medical research [103]. VH in Black communities is linked to medical distrust and structural racism [2] and given the course of medical history Black people have endured, they have reasons to be fearful and to be mistrustful [98].

Nephew [105], identified as a Black physician and experienced VH, “…experiences of structural racism overpowered a decade of medical education, if this experience of systemic racism cannot be understood then it will be difficult to understand VH in the Black community”. The historical mistreatment of Black patients within the healthcare system, medicine and research has caused a high prevalence of mistrust towards the COVID-19 vaccine [58]. A significant determinant of VH was historical mistrust due to the government, pharmaceutical companies, scientific research organizations, and medical establishments conducting unethical research and mistreatment in the Black population [61,62].

#### 3.4.7. Impact of Structural Racism

Structural racism helps to explain the lower rates of vaccine uptake in ACB communities [110] as it is the root cause of mistrust [59]. Distrust is a well-founded response to structural racism, structural inequities, and racism which remain problematic [57]. Low willingness to receive the COVID-19 vaccine predates the pandemic based on VH and low participation in clinical trials [108]. Furthermore, unvaccinated individuals were higher in cultural mistrust [71]. Black and White Americans’ experiences with the healthcare system are significantly different, and this may account for the different types of trust towards the government and institutions that are involved in vaccine development [98]. This has even caused some to wait to get the vaccine until authority figures, White patients, and the general population receive the vaccine first [58]. Mistrust was significantly associated with higher vaccine [27,69,99] and treatment hesitancy [27,69]. The success of vaccination uptake depends on the trustworthiness of the organizations offering the vaccine [111]. The least trusted included the government [60,63,64,67,75,92] particularly the federal government and the president [27], partly due to the politicization of the vaccine and its development [97], the conflicting guidance from the federal, state, and local governments, the government’s approach to healthcare [94], political meddling [62] as well as organizations that produce [67,78], market, and distribute vaccines, such as pharmaceutical companies [60,78], physicians [75] and healthcare providers [60], the healthcare system [63,92,93] and the medical community [92] which have contributed to varying degrees of VH from lowering confidence to unwillingness to resistance [67,75,92].

Increasing racial fairness is associated with increasing vaccine confidence and trust, including its perceived health benefits and recommendations to others [101]. However, overcoming racism is not mainly the responsibility of ACB people themselves [111]. COVID-19 presents an opportunity for the healthcare system to start making changes for historical injustices and discriminations [109] and eliminate disparities [98]. 

#### 3.4.8. Everyday Racism

Many articles cite historical racism as the source of mistrust in the healthcare system, but this ignores the everyday racism experienced by ACB people, such as being misdiagnosed, pain denial, and treatments being withheld [96]. In addition, continued acts of violence by police against Black people is a source of tension and mistrust with the government [61]. The political environment of promoting racial injustice is also a reason for VH [62]. Concerns about police violence were associated with concerns about the COVID-19 vaccine and worse mental health [71]. Willis et al. [85] reported that COVID-19 VH was 2.61× higher in those that reported discrimination by police or the courts, indicating a potential link between racial discrimination in the criminal justice system and VH in Black adults. Furthermore, Thorton & Reich [64] found that Black mothers experienced racial and bias and perceived rejection of vaccines as a form of resistance against governmental power. They also considered homeschooling as a legally acceptable way to avoid vaccinating their children since they were concerned about vaccines and the organizations that promoted them [64]. 

#### 3.4.9. Racial Burden

Racial burden refers to the undue hardships placed on the ACB population due to their perceived race. These analytical themes are comprised of the descriptive themes of concordance. 

#### 3.4.10. Concordance

A commonly proposed solution for the mistrust that has been brought up as a factor in low vaccine uptake in Black communities is positioning Black physicians and investigators at the forefront to provide racially concordant messaging and build trust [104]. This circumvents rather than addresses the issues of mistrust [104]. An individual is lived, and collective experiences of family, community, and history are not simply reflective of a Black patient-physician relationship, therefore a Black physician’s trustworthiness cannot be a substitution for that of the institution [104]. 

The concerns of mistreatment and distrust in the system are not irradicated by being a Black physician; VH still exists [106]. A Black women physician, researcher, and epidemiologist still felt uncertain after getting vaccinated after reviewing immunology, virology and trials and listening to interviews to understand the mRNA technology [105]. Black physicians are at a more significant burden than their non-Black counterparts as they earnestly work to encourage vaccination in Black communities while being underrepresented in medicine because of racism, this further generates systemic problems for those already overburdened and exacerbates racial inequities in the future [104]. 

#### 3.4.11. Systemic Barriers to Access

Systemic barriers to access refer to policies and practices ingrained in systems such as the healthcare system that impede the ability of people within the ACB population to access vaccines. 

#### 3.4.12. Systemic Barriers to Access (In General)

Low willingness to be vaccinated for COVID-19 may have pre-dated the pandemic based on low access, cost, and other related concerns [108] as flu vaccine uptake also continues to be low in the ACB community; attributed, at least in part, to environmental barriers to vaccine access [92,101] Structural and systemic barriers to vaccination stem from systemic racism and include transportation [59,92,103], distribution, availability, and access [59,103]. The racial inequities in vaccine rollout copy the disparities experienced throughout the pandemic, with fewer vaccines going to Black residents [104] and question how those distributing the vaccines to Black communities care about them based on historical medical mistreatment [94]. In addition, other perceived barriers to access include navigating the complex healthcare system and internet access [92]. Those with no internet could have difficulty getting relevant COVID-19 information or registering for appointments, as well as some elderly people may find it challenging to navigate [92]. 

Compared to other vaccines, there is a higher prevalence of mistrust about the equitable distribution of the COVID-19 vaccine [58]. Poorly coordinated responses between public health and the levels of government also created barriers [97]. Furthermore, healthcare leaders’ recommendations of mass vaccination of Black communities before the general public could lead to fears that they are being further tested [106]. 

Stemming from systemic racism, perceived barriers include medical mistrust, vaccines, healthcare providers, government, health systems, and pharmaceutical companies [59] due to historical mistreatment of Black patients within the healthcare system, medicine, and research [58]. Bias in medicine continues to influence vaccine uptake in Black populations [101].

### 3.5. Theme 2: Sentiments and Behaviors

Vaccine uptake on an individual level is influenced by sentiments and behaviors, which span from the willingness to get vaccinated, vaccine beliefs, lived experiences, and vaccine confidence in our findings. In addition, willingness to get vaccinated was involved in perceiving the vaccine with low necessity and reasons for increased willingness; vaccination beliefs included general overall beliefs about vaccines, as well as religious and political beliefs; lived experiences included mistreatment, death exposure, previous infection, previous vaccinations, and co-morbidities; and vaccine confidence included general distrust, the timeline and novelty of the vaccine, side effects, efficacy, and safety. 

#### 3.5.1. Willingness

##### Low Necessity

Those who were less willing to get vaccinated had low levels of vaccine necessity [61], had fewer general concerns about COVID-19 [75] and believed that other means of protection [92], such as wearing a mask, taking supplements, and staying home were more effective than the COVID-19 vaccine [58]. Whereas others did not take vaccines in general [91], were confident due to their current level of health [93], or felt that vaccinations were the wrong approach and that efforts should be focused on improving baseline health [62]. 

##### Increased Willingness

There is a variance in the likelihood of getting vaccinated for COVID-19 in the Black population [82]. Main reasons for participants not being VH is if they already received the vaccine, to protect themselves and others [93], such as family and small children [62], if it was mandatory for school or work [62,93], and for protection to feel safe and not worry of getting sick or dying from infection [91]. Willingness was also increased when there was the perception that other Black people were getting vaccinated and that it had social approval [67]. Vaccination also provided a greater perception of control from contracting COVID-19 for some [67], and those willing to vaccinate believe that the vaccine would not harm them [62]. Aubuchon et al. [67], also found that those with skepticism were more willing to vaccinate. 

#### 3.5.2. Vaccination Beliefs

##### General

Beliefs can mediate the association between race and vaccine intention; thus, the focus should move from intentions to beliefs regarding VH [86]. Furthermore, changes in a person’s perceptions of benefits, susceptibility, and seriousness of infection could influence the likelihood of vaccine uptake [80]; those who were more worried and perceived COVID-19 as dangerous, were more likely to get vaccinated [89].

##### Religion

Deep-rooted religious beliefs were associated with vaccine resistance [92], and some participants had religious concerns based on what is in the vaccine [89]. Sharma et al. [82] reported that those who followed a religion other than Christianity or atheism were reported as VH. 

##### Political 

Political affiliation contributes to VH [61]. Those who reported a Republican political affiliation were more VH [82]. However, the level of trust in US president Trump and public health agencies and officials was smaller in Black participants and was less consequential to their pro-vaccination beliefs [86]. 

#### 3.5.3. Lived Experience

##### Mistreatment

ACB people fear racially discriminatory treatment, and those who have had a negative experience with a healthcare provider (HCP) have increased VH [58,61,108], as ACB people feel that their lack of trust has been validated [61]. In addition, seeing others mistreated in healthcare, such as a child observing the mistreatment of a parent, may prevent them from seeking care themselves [102]. 

##### Death Exposure

Exposure to the death of a friend or close family member had a 1.70× higher rate of vaccine hesitancy, and further research is needed to determine if there is a causal connection and if it is specific to COVID-19-related deaths [85].

##### Previous Infection

Accepting the COVID-19 vaccine was not associated with a previous infection of COVID-19 in hemodialysis patients [108], and only 3 out of 10 Black patients who recovered from a COVID-19 infection were willing to accept a “safe and effective” COVID-19 vaccine [79]. 

##### Previous Vaccinations

Lim found that 66.1% of those with COVID-19 VH had received the flu vaccine, indicating a potential for vaccine receptivity. Rungkitwattanaku et al. [108] found that getting the flu vaccine was associated with willingness to get the COVID-19 vaccine. Although receiving the flu vaccine is a predictor for vaccination [10], flu vaccination rates remain low, with 1out of 3 Black participants 65 and older never having received the flu vaccine [68] and the low, consistent flu vaccination in Black women foreshadows the challenges of the COVID-19 vaccine [95]. 

##### Comorbidities 

Those with comorbidities felt that receiving the vaccine could put their immune system at risk [62]. Patients with congestive heart failure, coronary artery disease, diabetes mellitus, and hypertension had a higher likelihood of declining vaccination [79]. Those with lupus believed there was more potential for lupus flare-ups and lupus-related side effects, and it would have decreased efficacy [75]. However, Black participants living with HIV generally had high COVID-19 vaccine uptake; those who were more worried and perceived COVID-19 as dangerous were more likely to get vaccinated [89]. In addition, Cokley et al. [71], found the relationship partially mediated lower perceived discrimination between race-related COVID-19 vaccination concerns and mental health symptoms. 

#### 3.5.4. Vaccine Confidence

##### General Distrust

There is decreased willingness to get vaccinated based on mistrust of the vaccine itself [67,94] and a combination of personal fears and misinformation from the internet and peers [56]. Sekimitsu et al. [94], found that by the end of their study, a third of those who expressed VH had become vaccinated, potentially indicating fluidity in opinions regarding VH.

##### Vaccine Timeline/Novelty

The main reason for VH was uncertainty due to the COVID-19 vaccine’s quick development [62,91,93,102], the newness of the vaccine itself [61], and that more time [58,59,94] was needed for vaccine side effects and its efficacy [59] as well as to observe vaccine rollout [58].

##### Side Effects

Fear and concerns about vaccine long-and short-term side effects [61,94,95] was higher in unvaccinated participants [89] and those who declined vaccination [69,78,79].

##### Efficacy

Compared to other vaccines, there is a higher rate of mistrust regarding the COVID-19 vaccine’s efficacy [58]. Black people have expressed hesitancy and declined the vaccine due to uncertainty that it will be effective [78,92,102]. Those who expressed caution about the vaccine stated they would be more willing to be vaccinated after observing its effect in others [92], others stated that if the COVID-19 vaccine had 100% efficacy or if a sing dose was required, they would get vaccinated [89]. Concerns about vaccine efficacy were also the main reason for not getting the flu vaccine [95]. 

##### Safety

Black people continue to have low flu vaccine uptake due to concerns about its safety [95,101] and have expressed VH due to safety concerns for the COVID-19 vaccine [61,92,102], which has more mistrust about its safety compared to other vaccines [58]. 

### 3.6. Theme 3: Engagement and Influence

Engagement and influence relate to how those who interact with ACB people impact their health behaviors related to vaccine uptake. 

#### 3.6.1. Government and Healthcare Systems

Government and healthcare systems relate to how the political system and as well as the healthcare system impact ACB populations’ vaccination uptake. 

#### 3.6.2. Government and Healthcare Systems (In General)

Healthcare systems must mitigate factors that prevent making informed vaccination decisions [110]. Mistrust in the government and healthcare systems is evident today [63], and healthcare policymakers cannot overlook the mistrust and cynicism towards health professionals by those who are informed by historical events such as the Tuskagee experiments [110]. Government and public health officials need to rebuild relationships with Black communities [61,63], including committing resources to address historical factors [61] and addressing bioethical concerns relating to mistrust and COVID-19 VH [63]. The government and medical community need to work towards the quick and equitable distribution of vaccines in Black communities to build trust [105], including creating community programs that are transparent, and factual and have established relationships with community leaders, as well as follow-up with community members to ensure their needs are met [63]. The government also needs to advocate for programs and policies that help to elevate Black communities [61].

VH is associated with skepticism of the government and the healthcare system [56]; therefore, concerted efforts must be made by public health and the medical system to gain the trust of the ACB communities, which carry a higher burden of COVID-19 [91]. In addition, providers’ lack of cultural sensitivity, responsiveness, and competency in practice are also reasons for mistrust in providers and the health systems [59]. Health systems and organizations need to build rapport and trust in vulnerable populations to reduce disparities in vaccine uptake and reduce further COVID-19 related infections and deaths [76]. The health systems must be assessed for disparities and the drivers that effect change, as all factors are needed to develop effective intervention [77]. Health systems and professionals must intentionally seek to understand and address the fears inflamed by anti-vax leaders, implement interventions to combat negative media campaigns, and close knowledge gaps [77]. Healthcare organizations must also allocate funding to their public health departments and communities to dimmish disparities and build trust [77]. The most disadvantaged must be listened to, with reasons for distrust acknowledged, and transparency must be maintained to help bring about racial justice and restore trust [98]. Public health strategies must also concentrate on Black communities to address concerns and fears of racism involving the development and distribution of vaccines [97], such as community engagement, diversity in medical professionals, and transparency about COVID-19 vaccines [76]. Focusing on issues important to the community can help initiate possible vaccine solutions [107]. Black populations need long-term support to address their needs and concerns after vaccination [97]. 

#### 3.6.3. Pharmaceutical Companies

There are high levels of suspicion about pharmaceutical companies, furthered by the politicization of vaccine trials [111]. To help build trust pharmaceutical companies need to be transparent about delays and adverse effects of a vaccine but should let medical professionals and the public health officials communicate the safety and efficacy of the vaccine [98]. In addition, to further earn trust, the trials should have maximum transparency, ensuring all aspects have informed consent, that participants will receive medical care for any injury as a result of an experimental vaccine, and that Black communities will have fair access to the results, and that they will ensure that no vaccine will be submitted for approval unless thoroughly vetted for efficacy and safety [111].

#### 3.6.4. Healthcare Providers 

Healthcare providers relate to how those who provide healthcare to ACB populations impact vaccine uptake. Healthcare providers is a general term and include all professions that provide healthcare-related services to ACB people.

#### 3.6.5. Healthcare Providers (HCP) (In General)

VH in Black communities is historically justified, and Black communities should be vaccinated; therefore, medical professionals must do more to improve care and create a trust [103]. Service providers need to acknowledge wrongdoings and the failure to gain trust based on historical and everyday racism [105]. HCP and health agencies need to ethically consider the current drivers and effects of mistrust, to have adequate inclusion of vulnerable populations in research, to improve health literacy, and to define the roles of physicians in the health of Black Americans [63]. Relating the awareness of historical distrust in medicine and the acceptance of new medical interventions, such as COVID-19, can inform HCP efforts to provide excellent care and empower Black patients going forward [58]. There is a need to build trusted client-HCP relationships to educate and to recommend the COVID-19 vaccine to those experiencing VH [62]. These client-HCP relationships can help build trust decrease skepticism [63], and VH and improve confidence [93]. 

The vaccination decisions of Black patients were found to be based on discussions with their healthcare providers [58]. Providers should offer accurate, current information to high-risk Black patients about how to access vaccines [2] and engage and remind patients about the importance of vaccines [99], as well as target those who are living alone, isolated, and suffering from depression to reduce vaccine-related inequities [68]. Service providers also need to address the SDOH and act as partners in assessing, advocating, and implementing interventions for these disparities [102]. Scientists, researchers, and physicians must broaden their perspectives to understand factors that explain decisions and behaviors related to vaccination [110]. They also need to support patients in navigating the complexities of the healthcare system while emphasizing the importance of vaccination, including population-wide coverage, data on the number of Black clients that have been vaccinated, and highlighting Black scientists’ contributions [2]. 

Recommendations from a healthcare provider are a predictor of vaccination [10,60,61]. and direct recommendations by a medical provider may increase vaccination coverage in vulnerable populations [78]; however, although reassurance and recommendations from a trusted HCP can increase vaccination; a negative recommendation may influence them against getting vaccinated [62]. Providers need to be aware of their influence and consistently provide vaccine education and recommendations during medical consultations, such as the flu vaccine during visits with outpatient cardiology patients; vaccination rates were even higher when recommendations to cardiac patients came from their cardiologist, although most cardiac patients more often received vaccination information and recommendations from their family physician and internist, potentially due to frequency of visitation [10].

Physicians have a crucial role in positively influencing the vaccination behavior of their patients [10]; for example, older Black participants were more likely to get the flu vaccine if it was recommended by their physician [68]. In addition, healthcare providers could improve prevention measures, such as in the foreign Black population, by highlighting the need for eligible males to get vaccinated for HPV [70] and Black participants in a smoking cessation program mentioned a physician’s opinion as being influential to their decision to get a vaccination for COVID-19 [90]. Every physician is responsible for counselling each patient about vaccination while addressing their concerns [106], which includes having mutual and transparent conversations with their Black patients to build trust by listening to and acknowledging their concerns [100]. For example, Thornton & Reich [64] reported that vaccine-hesitant Black mothers view physicians as a potential threat with concerns about reporting their families to state agencies rather than as service providers or consultants privileged families do. 

#### 3.6.6. Pharmacists

Pharmacists can contribute to decreasing vaccine hesitancy and improving confidence [93]. For example, Schafer et al. [83], found that pharmacists partnering with a seniors’ center was an effective model for improving community engagement and supported pharmacists providing educational services.

#### 3.6.7. Racial Concordance/Benefits

Trust is critical to the health, and the lower vaccination intention of Black patients may be due to a general lack of trust toward the healthcare infrastructure where few medical providers look like them [96,106]. Vaccine transparency is only a short-term solution. Ultimately the Black community need healthcare workers who are from their neighbourhoods, represent their best interests, understand their experiences, and are as diverse as them [106]. It is known that Black patients prefer Black physicians, even waiting months for appointments [96]. Black healthcare providers have the unique role of providing for their patients as well as their communities [109]. Many participants reported that learning facts about COVID-19 were more impactful when hearing it from Black physician researchers involved in the vaccine developments [80]. 

Building trust in Black communities involves increasing the number of Black nurses, physicians, dentists, pharmacists, and allied professionals [103]. Black physicians and investigators need to be at the forefront of efforts for vaccine rollout, and public health messages need to come from Black health leaders and Black scientists; sharing their stories can be more directly relatable to the needs of communities [96]. Black men are underrepresented in the medical profession despite representation being critical to build trust in the Black community and advocating for Black people to get vaccinated [103]. More Black people also need to be recruited in research and vaccine trials [95,97] so that there will be equitable access to cures and increased community empowerment to ensure that the mistakes of the past are not repeated [95].

#### 3.6.8. Community

Community relates to ACB populations with similar characteristics or who live in the same general location and how they impact vaccine uptake.

#### 3.6.9. Engagement

Health promotion efforts should include community engagement [65] and dialogue to increase COVID-19 vaccination acceptance since disparities in racial immunization are likely due to structural racism [84] and culturally competent outreach, such as townhalls, could be used to answer questions and counter misinformation [57]. Healthcare providers should link with the community through communication channels, including community outreach and social media [57]. Findings suggest that reception for COVID-19 public health messages may be increased by healthcare providers and through community-based and non-political entities [27]. Outreach should also be led by community leaders and peers, with a focus on Black people who are young, depressed, and those with low socioeconomic status [75]. In addition, grassroots organizations can engage Black communities personally and culturally sensitively [109]. Community engagement is also needed for accurate messaging and behavioral change [65] and helps to improve transparency; it is an internal asset-based approach rather than an external top-down problem-solving approach [97].

#### 3.6.10. Collaborations

To help increase vaccine uptake, there is a need to increase communications and collaborations between healthcare providers, Black people, and the government [61]. Collaborations need to be bidirectional [111] with round table discussions [100] and learning that involves grassroots organizations and individuals that are trusted by the Black community members and leaders [111], to address concerns [89] including those about health equity and antiracism initiatives to increase vaccine uptake [100]. VH is changed with knowledge and situational context, through collaborating with communities to establish community-based clinics and by employing navigators and coordinators, accessibility issues can be reduced [92]. 

#### 3.6.11. Partners 

HCP collaborating with partners that patients trust can also build trust and help to identify the best alternatives to improve the quality of care in the population, including creating targeted messaging around COVID-19 [108]. In addition, community-academic partnerships and collaborations can help reach Black communities and reduce VH on high-risk groups, as long-standing relationships with the community may strengthen intervention initiatives [80]. Furthermore, community leaders and educational committees should create equity and anti-racism initiatives, including messaging that targets vulnerable populations, including misinformation and vaccine availability [100]. Partnerships with community members, church leaders [61] and faith-based organizations [109], local government [61] and trusted sources such as political advocacy groups [109] can increase community capacity by co-creating solutions including public health messaging to increase trust [61] and vaccine uptake [61,109]. In addition, Black led partnerships between stakeholders and healthcare can increase [57] and improve outreach as well as address anti-Black racism to increase vaccine confidence [2] Black HCP including physicians and clinicians are trusted messengers and should work in the community [57,110]. Working together with HCP and community-based advocacy groups can also help dispel misconceptions and build trust [79]. Increases in COVID-19 vaccine uptake in Black Americans have been attributed to the successful outreach of health leaders, medics, faith organizations, and community-based organizations reaching to Black communities, including through live stream town halls [99].

#### 3.6.12. Relationships

Relationships must build and establish stronger collaborations for COVID-19 and future interventions for lasting and impactful change [107]. In addition, the healthcare industry must directly engage in building authentic relationships as vulnerable communities continue to be impacted, such as by the potentially lethal combination of COVID-19 and influenza “twindemic” [95]. 

#### 3.6.13. Community Mobilization

Community mobilization relates to how the community as a whole can engage to impact low vaccine uptake in ACB communities.

#### 3.6.14. Community Mobilization

Black communities must mobilize to provide evidence-based information [101]; including engaging in research and community workers providing information on vaccine safety and efficacy to places where they gather such as churches and recreational areas [110], to support vaccine acceptance [101]. Community support initiatives such as sharing testimonials to motivate others, can also decrease VH; for example, the first American to receive the COVID-19 vaccine was a Black intensive care nurse hoping to inspire other Black people who were skeptical about getting vaccinated [99]. Testimonials from trusted professionals and HCP from the community can act as frontline advocates [92]. 

In addition, community-based wellness programs can enroll community members for vaccination, organize clinics in churches and connect community members to healthcare providers [61].

#### 3.6.15. Community Leaders

Community leaders as people within the ACB community that are respected and can influence others in the community. 

#### 3.6.16. Community Leaders (In General)

Community leaders and influencers need to be identified to advocate in Black communities [62]. Black community leaders should be involved in all steps of vaccine development, distribution, and monitoring to address VH [97]. 

#### 3.6.17. Church 

The behaviors and attitudes of church officials have the social capital and influence needed to promote vaccinations, such as increasing the likelihood of accepting the COVID-19 vaccine by Black adolescents living in rural areas [56]. In addition, a community faith-based health and wellness program can be a trusted source of information about COVID-19 vaccinations in the Black community [61].

#### 3.6.18. Social Interactions

Social interactions include various forms of communication with others that influence healthcare-related decisions. 

#### 3.6.19. Peers/Family

Black parents often act as buffers and filters for their child due to social dispositions and mistrust that have been racially socialized from integrated trauma and experiences and provide specific practices and messaging related to racial identity [102]. The behaviours and attitudes of older family members also influence the likelihood of Black adolescents accepting the COVID-19 vaccine [56]. Furthermore, vaccine hesitancy may be conflated with diligence; for example, women are often tasked with caring for others at low wages and with little protection, yet Black women are also the most vulnerable, so it is prudent to question vaccine safety and efficacy [112]. 

Peer relationships, including family, can be a source of misinformation [73,102] and can increase susceptibility to conspiracy theories, such as the vaccine being created with malicious intent against Black people, and has been associated with VH [102]. However, Francis et al. [73] found that those that discussed vaccination with their families had more positive expectancy and more positive injunctive norms. Therefore, promoting positive conversations and helping to develop family and network-based interventions may help to create positive COVID-19 vaccination beliefs and decisions [73].

Peer influence does make a difference in increasing willingness to get vaccinated [94,99]; in college-age participants, vaccine acceptance was related to peer pressure and fear fatigue over worrying about themselves and others [92]. In addition, factors that included being able to go out to events with friends and family also increased the willingness to get vaccinated [94]. 

#### 3.6.20. Social Norms and Context

The attitudes of social networks can influence and encourage vaccination [69]. The success of the COVID-19 vaccine is dependant on if the members of ACB communities trust that it is safe; both perceived social approval and the perception that other Black people are getting vaccinated have been shown to increase willingness to get vaccinated [67,111]. It has also been found that weak subjective norms for a close social network were a predictor for not wanting to get vaccinated [69]. The social norms of different groups independently and interactively affect Black American’s vaccination intentions; for example, the social norms that exist in the general population (all Americans) were not the basis for decisions regarding COVID-19 vaccinations in Black Americans. However, the social norms of an ethnic group did predict vaccination intention [87].

Communal safety, as well as the perceived risk of infection, influence vaccination intentions [92]. Peteet et al. [80] found few participants concerned about hospitalization despite personally knowing someone who had contracted COVID-19, and Black Americans who were VH from the beginning may have become even more resistant due to perceiving it to be unnecessary, as those with lower vaccine intention had higher potential social norms and may have felt a false sense of protection from perceived herd immunity [87]. 

Witnessing the safe administration of the vaccine to social leaders, including healthcare professionals and politicians, could convey trust and show that the vaccine is administered safely for their benefit [67,106]; however, Sekimitsu et al. [94] found that seeing friends and family vaccinated but not celebrities or politicians would influence ACB populations to get vaccinated.

In addition, Momplaisir et al. [62], found that participants believed that there would be extremely low vaccine uptake in their social network and that a famous person from the Black community to serve as a role model would not sway their decision. Furthermore, the social context within which COVID-19 vaccinations occurred must also be considered; vaccine uptake has been influenced by Black Lives Matter and the impact of the pandemic itself, which caused disruptions at work and at school and further increased anxieties [102].

### 3.7. Theme 4: Knowledge and Communication

This main theme describes the knowledge about vaccines and how information is communicated. Information about vaccines spanned from mentioning lack of information; race-based data; misinformation, myths, misconceptions, and infodemic or being overwhelmed by too much information. Articles also described information sources, including the most used, least trusted and most trusted sources. Communication was described through open dialogue, and messaging. Trusted messengers, message dissemination and interventions and strategies to improve vaccine uptake. The scope of messaging was further broadened by explaining the diversity, clarity, the need to acknowledge underlying issues that contribute to low vaccine uptake, transparency, providing more information, creating tailored messaging, and the effects of persuasive messaging. Interventions and strategies were further broadened in the literature by describing build trust, policies, addressing disparities, creating standing orders/protocols for vaccination, and specific initiatives taken in research. 

#### 3.7.1. Information

Information is data that is obtained to increase knowledge on vaccination-related topics. Although evidenced-based facts are desired, inaccurate data can also be disseminated to influence healthcare behaviors. 

#### 3.7.2. Lack of Information

Knowledge and awareness about vaccines influences VH [61]. Black parents are actively seeking information on childhood immunizations, including the safety of vaccines and their pros and cons [77]. Lack of information is the main contributor to VH [93], with limited data on the short-and-long-term effects being reasons for VH [62,89]. 

Not having enough studies and not knowing what was in the vaccine were the most common reasons for not being willing to take the COVID-19 vaccine, even if freely available [91]. Those with low HPV acceptability stated concern about the unfamiliarity of the vaccine and wanted more information [60]. Information on vaccine efficacy, side effects, safety [59,90] and the initial outcome of others would be help make decisions [90]. In addition, those familiar with vaccine immunology were hopeful that with more information, people would be more willing to get vaccinated [106]. 

#### 3.7.3. Race-Based Information

Black people need to be represented in the research as generalizations about ethnic minority groups can result in unfair distributions of research burdens and benefits [63]. More qualitative data is also needed on the lived experiences of Black people during the pandemic to improve experiences, access, and health outcomes [2], and to identify and dismantle barriers [2,101] to increase vaccine acceptance as well as overcome historical mistrust in clinical trials [101].

#### 3.7.4. Misinformation/Myths/Misconception/Infodemic

Misinformation and mistrust are the main causes of VH in Black communities [65]. Misinformation, lacking clear, culturally relevant information, and timely and accurate race-based data collection have contributed to mistrust [97]. Social media misinformation contributes to VH [61] and a media review also suggests that anti-vaccination leaders target the Black community with misinformation and skepticism [77]. In addition, the online COVID-19 misinformation that targets Black communities cannot always have their sources determined and may be people masquerading as Black people [102]. Misinformation campaigns that targeted Black communities caused a plateau in vaccine uptake after behaviors began to shift from being against vaccination to considering it, increases in vaccine uptake began to increase again as people observed pain and suffering brought from the new delta variant, and new vaccine mandates were put in place [99]. In addition, VH in Black communities due to misinformation is indicative of literacy gaps [2]; the infodemic also exacerbated poor health literacy and contributed to VH [61]. To increase vaccine uptake, education is needed to combat the misinformation [103,108] spread on social media, as it can influence decision-making and hesitancy [103]. 

Unvaccinated participants were also concerned about what was in the vaccine, such as microchips and materials from pigs or fetuses, as well as irreversibility, altering DNA, medical history, and bioweapon technology [89], and getting infected from the vaccine [62]; conspiracy theories and myths are factors associated with vaccine resistance [92]. In addition, low vaccine intention may be caused by misconstrued herd immunity and a false sense of protection against COVID-19 due to high societal rates of vaccination and infection [87] and perceived immunity against COVID-19 re-infection [79]. 

#### 3.7.5. Information Sources

Information sources pertain to where vaccination-related data is obtained. 

##### Most Used Sources

Physicians were the most frequently used sources for information, and other sources were television, newspapers, church, virtual town halls, hospital websites, family and friends, and a few participants stated they used social media [94].

##### Least Trusted

In addition, the government [94] and specifically the federal government and president were the least trusted sources, followed by social media [27]; the news and support and advocacy groups were also found to be less trustworthy [75]. 

##### Trusted Sources

Trust in vaccine information increases willingness to vaccinate [67,86]. Encouragement from trusted individuals may change people’s minds about becoming vaccinated [93]. Trusted sources could potentially be used to communicate vaccine information and mitigate the effects of misinformation [86]. The most trusted sources of information were healthcare providers and health professionals, followed by health officials and health agencies and finally, local government officials [27], including Black doctors, researchers, and trusted leaders [59]. Olanipekun et al. [78], found that 8% of participants were willing to accept the vaccine based on information from TV, radio, or internet ads, whereas 75% were more willing to accept the vaccine if it was recommended by their primary care physician or specialist; however, Woko et al. [86] found that Black Americans were more likely to trust social media. Those who obtained their information from less reliable sources had a higher likelihood of misinformation which led to higher levels of VH, whereas those that obtained information through physicians and professionals had a better understanding [65]. Although vaccination beliefs are associated with trust in information sources, they do not account for the differences in vaccination beliefs by race [86]. 

#### 3.7.6. Open Dialogue

Open dialogue refers to a style of communication whereby participants can speak freely without the constraints of closed ended questioning. 

##### Open Dialogue

Public health campaigns should include having open dialogues with trusted and credible scientists and healthcare providers [59]. Discussions should include full disclosure of results to make an informed choice and potentially persuade vaccination [89]. In addition, using a communication framework may help clinicians bridge gaps and improve vaccine uptake [2].

##### Messaging 

Messaging is a form of communication with a specific purpose and or audience; messaging is provided to increase vaccine uptake in ACB communities. 

##### Diversity 

Visuals and narratives used in vaccination messaging should have diversity across race, age, and gender (HPV) [60].

##### Clarity

Vaccination messaging must use straightforward language with precise and lay language on safety and efficacy [27,103]. Communications must be comprehensive to the audience as well as appropriately acknowledge uncertainty [98].

##### Acknowledgement

Acknowledging and addressing mistrust can increase equitable vaccine access by improving vaccine confidence and vaccination intentions and rates in Black communities [59]. It is important not to blame the individual for VH but rather to acknowledge the past and present injustices rooted in policies [109], including historical medical mistreatment and health disparities resulting from unconscious bias and racism [100]. 

Systemic racism must be acknowledged as a mistrust’s root cause [59]. Vaccination promotion strategies need to acknowledge systemic racism as the fundamental cause of mistrust [59], address the abrasive relationships between the Black community and the police [61], and acknowledge concerns. Rather than implying that the lack of trust in HCP means ACB people have something wrong with them or that their health-related problems wouldn’t exist if they trusted more, it is important to acknowledge the roles of the health institutions in creating distrust and to recognize that the role of trust should not be overvalued, as it is part of the conversation but not all of it [112]. 

##### Transparency

To promote vaccine uptake and combat mistrust, messaging must be transparent and be maintained to help bring about racial justice and restore trust [98] and to increase understanding of health-related information [103]. Preferred messaging includes transparency about side effects [59,60,62,97], risks and benefits [60,109], development [62,65], safety [62,69], eligibility [60], and efficacy [62,69,97], In addition, information needs to be provided on specific groups, such as children and pregnant women [97], and vaccine specific information such as cancer prevention for HPV [60]. 

Due to the rapid development of the COVID-19 vaccine, questions must be answered promptly with full disclosure to earn people’s trust [106]. It must also be stated that the vaccine is not being tested on vulnerable populations; but safely administered for their benefit [106]. Furthermore, misinformation and conspiracy theorists that exploit historical injustices must also be addressed [97]. along with historic and experienced-based mistrust [57]. Misconceptions about COVID-19, such as safety and side effects, raise ethical questions about health literacy levels in Black communities and how to improve them [63].

#### 3.7.7. More Information

More data is an important intervention to increase willingness to get vaccinated, such as more information about the people who have taken the vaccine [94]. Not having enough information about vaccine safety and efficacy has increased VH [108]. More information is needed to inform Black communities [103] and rather than pressuring an individual to be vaccinated, the focus should be educating the population [106]. Messaging should also include information about trusted sources [59], resources for additional information [60], and available community support [65] and emphasize the protection of friends and family [94]. 

#### 3.7.8. Tailored Messaging

Tailoring messages for specific groups and demographics [65,72,108], such as age, could make information more concise and promote vaccination in Black communities [65], including Black immigrants [70] and underserved older Black people [68]. Messaging needs to be informative and educational, while addressing apprehensions on safety and efficacy in culturally sensitive way [109].

Culturally tailored messaging with the faces and voices of the intended Black audience could build trust in the community through representation in promotional materials [60]. In addition, tailoring messaging could reduce skepticism among Black adolescents as well as address misinformation related to side effects and governmental distrust [56]. An Afrocentric approach which centers on the clients’ values and perspectives would also improve vaccine uptake [2]; including acknowledging the past unethical treatment of Black people in research [72], acknowledging current and historic racism and discrimination [69], emphasizing current safeguards to prevent mistreatment, and stating the roles of vaccines in reducing racial inequities [72].

Public health messaging [72], as well as patient education [10], should be tailored to specific vaccine concerns [10,72] to improve vaccination uptake in vulnerable populations, particularly in those with heart failure and other high-risk conditions [10]; personalized medical advice from a physician would increase willingness to get vaccinated [94]. People’s perceptions of vaccine efficacy should also be assessed before message exposure, so that tailored messages do not create reactance of message rejections [81]. Furthermore, comparing and contrasting racial differences in COVID-19 vaccination rates and interventions should be used with caution, when using the social norms of all Americans, to mobilize Black Americans [87]. 

#### 3.7.9. Persuasive Messaging

Narrative messaging and self-persuasion narratives were associated with a greater vaccination intention. Self-persuasion narratives also had more positive vaccine beliefs with higher vaccination intention; it would be beneficial for a mass media campaign to include stories about how people change their minds about the COVID-19 vaccine [74]. In addition, it is essential to understand the relationship between individual perceptions, message framing and reactance in the context of vaccination, particularly among the Black population, which is disproportionately affected [81]. Richards found that using loss-framed messaging with parents that had a low perception of the efficacy of the HPV vaccine were more reluctant to vaccinate their children than those who viewed gain-framed messaging; defense-motivated process and psychological reluctance occur with loss-frame messaging, and it was less persuasive than gain-framed messaging. 

#### 3.7.10. Trusted Messengers

Trusted messengers have established a trusting rapport with ACB communities to work with them to improve vaccine uptake. 

#### 3.7.11. Trusted Messengers

Trusted messengers help to gain the trust of the Black community, which is essential for health communication, by disseminating accurate information and promoting vaccination behaviors [65]. Trusted messengers within a college environment include student leaders, coaches, and faculty who can deliver vaccine-related messages to Black students, use social events such as homecoming and football games to reach targeted populations and conduct health communication campaigns through open dialogue with stakeholders [65]. Pastors are also crucial trusted messengers within the Black community, and others in the faith community, they can work with governments and institutions to facilitate discussions, provide information, develop measurable improvements and help build trust [107]. In addition, Black HCP are trusted messengers [57,110] and their vaccine endorsement, along with other community leaders, peers, and scientist, can also increase willingness to vaccinate [92]. 

#### 3.7.12. Message Dissemination

Message dissemination is the process of dispersing evidence-based vaccination information to reach ACB communities. 

##### Message Dissemination

Effective vaccine message dissemination includes physical location, word of mouth, social media [60], and other virtual tools which can reach large audiences, especially during the pandemic when in-person social interactions were restricted [80]. Information can be provided by answering questions online and at physical locations such as mosques, barbershops and other trusted community-based organizations [103]. In addition, timely information in multiple formats should be brought to community vaccination sites such as churches and community centers [92]. 

To increase trust and confidence in Black communities a combination of key messengers including healthcare and community leaders [106], social events [65], multi-source social media [65,106], and other trusted channels should be used [57]. Furthermore, mobile vaccination programs can also rapidly disseminate vaccine promotion messaging [101].

##### Interventions/Strategies

Interventions and strategies refer to the way to improve vaccine uptake in ACB communities. 

##### Building Trust

To increase vaccine uptake, novel initiatives that build trust and support evidence-based medicine need to be established [110], including ensuring protection from unethical research practices and screening and treatment strategies for chronic depression, disabling and less likely to be treated in Black people. Trust impacts the intention to get vaccinated, and increased medical trust is associated with an increased willingness to be vaccinated [84]. Lack of trust in the vaccine is one of the most common reasons for not being willing to take the COVID-19 vaccine if freely available [91]. Privor-Dumm & King [107] proposed a framework to build trust and acceptance, which includes understanding history and contexts, listening and empathy, engaging pastors as trusted messengers, creating partnerships with shared responsibility and power, and co-creation of solutions with faith leaders and their community, governments and institutions to create sustainable, long-term change. 

##### Policies

Policy initiatives are needed to build trust in the health benefits of vaccines in Black communities [101]. Making vaccines mandatory for travel and giving a choice of vaccine [89], as well as for school or work [62,93], would persuade people to get vaccinated [62,89,93]. Burki [99] stated that policies should require all populations to be vaccinated for public health and Okoro et al. [92] found that mandatory vaccination influenced vaccine acceptance; however, Thornton & Reich [64], stated that low vaccine uptake may be due to an intentional refusal and parental agency rather than structural barriers, so requiring vaccination in children’s programs could alienate families [64].

##### Addressing Disparities

Racial/ethnic disparities must be addressed and eliminated to support health equity [101]. Healthcare should be viewed as a means of preventing the SDOH that worsens health outcomes [95]. In addition, by prioritizing vaccine acceptance in communities that are the most susceptible to COVID-19 infections, the protection of the larger population and individuals nationwide is also increased [94]. 

Culturally responsive strategies also need to be developed as individuals may lack the personal agency to address racism and everyday discrimination that are structural barriers to finding, accessing, and receiving equitable health services [71]. Low continuity of care, care satisfaction, and quality of care are associated with a decreased likelihood of getting vaccinated [68]. Furthermore, there is a need to maximize vaccine availability and streamline the process [59], as the lack of access to COVID-19 related healthcare and vaccines has increased distrust [64]. 

Accessibility should be increased by placing Black people on a vaccine priority list or providing them with greater vaccine access [2]. Furthermore, accessibility should include ensuring that there are multiple medical and non-medical access points [59], such as placing vaccination centers where people want to be vaccinated [2] within the community, that are trusted and convenient like barbershops [57,99], churches [57] and hair salons. Initiative should include assisting neighborhood and mobile vaccination distributions [100], improving access to hard-to-reach communities [101], as well as helping to improve health equity to counter racism [100]. This could be especially helpful for lower socioeconomic groups [99] and rural communities [97] that often need to travel further to be vaccinated [99], which can be timely and costly [97]. Cost could also be mitigated by improving health insurance, particularly for Black immigrants, with low vaccination rates; health insurance remains crucial for HPV vaccinations [70], and accessibility should not be cost-prohibitive [98]. However, providing a monetary incentive for vaccination was an ineffective strategy as in increased suspicions [65]. 

Unified service providers can help booster mobile infrastructure [101], including building equitable mechanisms to address VH through genuine communication, accountability for vaccine delivery, thoughtful partnerships, relevant messaging, and focusing on removing barriers to vaccine uptake [109]. 

##### Standing Orders/Protocols

Based on findings it is recommended that cardiology and primary care clinic leadership have standing orders and protocols in the electronic medical record system, to allow healthcare providers such as nurses to recommend flu vaccines and vaccinate patients without a direct physician order or supervision unless an assessment is needed for a genuine medical contraindication [10]. 

##### Initiatives

Intervention initiatives to promote vaccination should focus on building trust [57,65], and misinformation, improving vaccine access [57], and including robust education about the health benefits of vaccines in Black communities [101]. In addition, HCP should be assessed for cultural competence and training on incorporating culturally based healing practices and building trust [59,105,109,110], with a focus on the SDOH for health equity [109].

Schafer et al. [83] initiated an education program to improve beliefs about pharmacists, pharmacies, vaccinations, and pneumococcal disease, and found improved vaccine uptake initially, but returned to baseline after 3 months; trust in pharmacists also improved initially but dropped after 3 months. Findings indicate the need for sustained efforts and that healthcare encounters can impact beliefs [83]. 

Abdul-Mutakabbir et al. [66] initiated a multi-tiered community approach to be effective, whereby faith leaders engaged with the academic community to disseminate COVID-19 information, culturally representative healthcare professionals delivered educational webinars, and used low barrier access sites, to target back communities; Peteet et al. [80], also found an increase in willingness to get vaccinated after a webinar. 

A tablet-based approach to improve vaccine uptake by Budhwani et al. [88] currently underway, uses tailored messages for youths within a rural context, motivational interviewing and text reminders to emphasize youth autonomy and decision-making, as well as to reduce healthcare provider burden. 

##### Gaps in the Literature

The gaps that were found within the literature included knowledge about how historical racism can be effectively addressed in a manner that can build/rebuild trust in our current healthcare system; defining why racism and discrimination in the healthcare system still currently exist; how to support racially concordant professionals, particularly their mental wellbeing; how to support the responsibilities of effective vaccination communication to build a trusting relationship during times of dynamic changed; effective methods to increase ACB participants in vaccine trials or methods to increase racially concordant providers to serve ACB clients; and how to improve health literacy in the ACB community to combat misinformation, targeted information with malicious intent and the infodemic. 

## 4. Discussion

This scoping review (ScR) sought to find the breadth of current evidence sources on low vaccine uptake in African, Caribbean, and Black (ACB) populations relative to public health in high-income countries (where they were considered a minority in comparison to the countries’ general population). Through this exploration, the main concepts found within the 60 articles examined included inequities and racism, sentiments and behaviors, knowledge and communication, and engagement and influence. Further details of these main themes were shown through their coinciding analytical and descriptive themes to reveal the breadth of findings while defining conceptual boundaries within the literature. By defining these concepts, gaps were revealed that warrant further research and commitment to further understand low vaccine uptake in ACB communities. 

Theory-based assessments help to improve the understanding and processes involved in creating effective interventions while aiding in the systematic identification of barriers and facilitators that can guide the designing, implementation, and evaluation of strategies [113]. The socioeconomic model (SEM) developed by Bonfenbrenner [114] will be used in this discussion to organize the social structures that influence behaviors at the individual (micro), community (meso), and systemic (macro) levels; these have been found to be relevant in addressing health inequities among racialized communities, particularly on data about the social determinants of health (SDOH) and addressing health inequities [49]. In addition, the intersectionality approach will be used to further understandings about how power domains within cultures, disciplines, systems, and interpersonally, cause people’s lives to be disadvantaged or advantaged [89]. Social justice is central to intersectionality, which considers the larger social contexts that influence experiences and daily interactions [115]. Therefore, the intersectionality approach will be used to further understand how mutual relationships between preparedness, response, and future alternatives to healthcare needs are impacted by the dynamics of power and the principles of human rights [116]. 

### 4.1. Macro/Systemic

The main concept of inequities and racism ranged from inequities, roots of distrust, racial burden, and systemic barriers to access. Inequities was further defined by the exacerbation of vaccine hesitancy (VH) and the social determinants of health (SDOH); roots of distress ranged from historical mistreatment, the impact of structural and everyday racism; and racial burden focused on issues of racial concordance within healthcare. In addition, the main themes of engagement and influence also described the impact of government and healthcare systems on vaccine uptake in ACB communities. The gaps that were found within the literature included knowledge about how historical racism can be effectively addressed in a manner that can build/rebuild trust in our current healthcare system, as well as not defining why racism and discrimination in the healthcare system still currently exist. These gaps are problematic given that health equity issues persist despite some decade-long commitment of public health towards health equity efforts [117]. 

Historically based structural racism has been the foundation for policies that have contributed to the racial health inequities that have been highlighted by the COVID-19 pandemic [118]; this must be considered to effectively implement change, as policies and power can affect the development and well-being of people by determining the circumstances in which they live [114]. Inequalities exacerbate what has been referred to as VH in the literature; however, our results show that low vaccine uptake is not solely based on sentiments such as hesitancy but also on systemic barriers such as living in areas with high socioeconomic vulnerabilities and having a lower level of education. These barriers perpetuate health disparities, which then manifest as low vaccine uptake in ACB communities. In addition, results show that low vaccine uptake is also rooted in distrust that has been in part brought about through historical mistreatment and racism, including stories of medical experiments that have been past down through generations and resulted in distrust throughout the healthcare system, which is further compounded by everyday racism such as a political environment that promotes racial injustices and bias. We must seek to understand low vaccine uptake as a manifestation of systemic and historical racism, as it has already been well-documented that socioeconomic and racial disparities in health status, health outcomes, and access to healthcare services are found in neighbourhoods with higher concentrations of racial minorities and socioeconomic disadvantages and that COVID-19 has disproportionately impacted ACB communities for numerous intersecting factors such as the increased risk of exposure to COVID-19 due to being over-represented in front-line work, more likely to use public transit, and less likely to have access to appropriate personal protective equipment [40,119,120]. Unfortunately, with reports such as by Thorton & Reich [64] of parents wanting to remove their children from school to avoid vaccination mandates, the historical and intergenerational impacts are being perpetuated into future generations; this dynamic should be a focus of future interventions to stem exponential consequences that historical misdeeds and unethical practices have had on ACB populations. 

Another consequence of the impacts of structural racism and the resultant mis/distrust is the adoption of a “wait and see” approach that has contributed to low vaccine uptake in ACB populations. These acts highlight the willingness of becoming vaccinated, but only after they can assess the impacts on others; the consequences of racism in this regard are succinctly articulated in the title of an article by Hoffman [121], “I won’t be used as a guinea pig for white people”. Statements such as these may indicate stress, and anxieties felt within ACB populations, whereby the fear is in the vaccination process, such as who created it and their testing and dissemination practices. 

Efforts to mitigate low vaccine uptake in ACB populations must not come at the expense of the rights and well-being of others. The use of racially concordant professionals to sway feelings of mistrust/distrust may exacerbate the racial burden faced by concordant professionals; not only may they have their concerns and feelings of mistrust, based on historical mistreatment and everyday racism, both inside and outside of the healthcare system, but they are being asked to speak on behalf of those that have perpetuated the racism and discriminations. The American Medical Association’s (AMA) report [122] found that during the COVID-19 pandemic, Black physicians experienced an increase in racist treatment from both patients and colleagues, which was supported by institutional structures. Furthermore, they experienced burnout, poor mental well-being, career dissatisfaction, contemplation of career change, and higher job turnover [122]. There is a gap in the literature on how to support racially concordant healthcare professionals in their work and for their own physical and mental well-being. It is unreasonable to ask racially concordant professionals to tell ACB communities to trust a system in which they experience racism; including being overburdened due to being underrepresented in the healthcare system [104]. 

Results show that not only has historical and everyday racism contributed to low vaccine uptake in ACB communities, but structural barriers created through systemic inequities have influenced vaccine distribution, availability and access, such as lack of transportation and internet access, the perception of poorly coordinated efforts between public health and the various levels of government, and the inability to navigate the complexities of the health care system. Structural barriers disproportionately affect Black populations, and strategies must demonstrate credibility in healthcare systems and acknowledge that issues about vaccine access are founded in structural racism [123]. Racism goes beyond conscious actions; it is so deeply rooted in our systems in Canada that it can be missed [43]. Low vaccine uptake is a systemic problem, and therefore it requires systemically based solutions. Government decisions must be based on human rights principles and be informed by consistent engagements in communities for inclusive and diverse feedback and representation; by understanding the issues decisions can be guided by equity and human rights to form sustainable interventions [116].

The main theme of engagement and influence, the impacts of government and healthcare systems on low vaccine uptake in ACB populations included the need to address not only health disparities but also their causes. Healthcare policies must acknowledge the mistrust that has been established based on historical events of unethical medical mistreatment (such as Tuskegee) and seeks to understand how “anti-vax” campaigns have inflamed fears of mistreatment. In addition, it is important to recognize that the SDOH, including racism, requires politically based solutions with scientific underpinnings and a focus on upstream interventions that take responsibility and ensure that universal measures are not at the expense of racial health equity, i.e., a race-neutral approach can further the divide [124]. Furthermore, using an intersectional approach can be used to design policies that consider differences between individuals and groups to mitigate disparities rather than increasing them [116]; for example, although social distancing has had socioeconomic consequences for society in general, it has more negatively impacted vulnerable populations including racial/ethnic minorities, those with low socioeconomic status, women, and children [116]. There must be concerted efforts to build trust, counter racism, and empower ACB communities. Furthermore, result show government and healthcare systems must also recognize that cultural incompetence of providers negatively influences how they are perceived; therefore, efforts should be made to improve cultural competencies within healthcare organizations and providers. Additionally, strategies to improve vaccine uptake should include implementing long-term support for communities, seeking to diversify service provider further, and ensuring transparency throughout the vaccination process to build trust. 

In addition, the main theme of knowledge and communication showed that a failure to collect quality race-based data could result in generalizations that contribute to inequities. However, there remain gaps in the literature on how to collect, analyze effectively, and translate race-based data to assess how intersecting factors impact individuals and communities as well as how systemic, and structural inequities contribute to socioeconomic status and the disproportionate effects of the COVID-19 pandemic [40]. Race-based information was also shown to help to identify and dismantle barriers, build trust, and increase vaccine uptake. The lack of available race-based data in Canada has contributed to the failure to effectively address racial health inequities and is a form of systemic racism [120]. The scarcity of data on intersecting factors such as occupation and ethnicity has been an obstacle in the identification and explanation of the overrepresentation of COVID-19-related deaths in ethnic minorities [40]; the collection of race-based data allowed for the broad recognition of low vaccine uptake in ACB populations. 

### 4.2. Meso/Organizational

How ACB people feel and behave regarding vaccination is influenced by the communities in which they live and the organizations responsible for their health and well-being. Furthermore, government initiatives such as the “Supporting Black Canadian Communities Initiative” exemplify how macro level systems can impact organizations [43]. The central theme of engagement and influence ranged from government and health systems, healthcare providers, community, community mobilization, and community leaders, to social interactions. Although government and healthcare systems were primarily described in the literature on the macro level, the description subtheme of pharmaceutical companies was described on a meso level. Additionally, healthcare providers included pharmacists and racial concordance benefits; community included collaboration, engagement, partners, and relationships; community leaders included the church, and social interactions included peers and family as well as social norms and context. 

Although healthcare-related organizations are within the more extensive healthcare system, findings show they can also take initiatives to build rapport and trust within ACB communities, including funding departments that seek to mitigate health disparities. The high level of suspicion of pharmaceutical companies in ACB communities is shown in our results. This is consistent with a report by the Action Center on Race and Economy [125] titled “Poison how big pharma’s racist price gouging kills black and brown folks” shows an example of current issues that establish trust in the organizations that produce the vaccines difficult. This report states that pre-existing and socioeconomically based health conditions, such as diabetes and hypertension, are found at higher rates than in White Americans and that the high cost of prescription drugs contributes to financial strains and thus perpetuates the cycle of health inequities. Our results show that to help address the high level of suspicion towards pharmaceutical companies; they must combat the politicization of vaccine trials, be transparent about the vaccine throughout the process, have trusted health professionals communicate vaccine safety and efficacy, take responsibility for the health and safety of trial participants, and thorough vet vaccines before applying for approval. Although pharmaceutical companies issued a joint statement that they would not use emergency authorization or submit a COVID-19 vaccine for approval that was not shown to be safe and effective through a phase 3 clinical study, it did not commit to the transparency of data or rule out the submission of an application before the end of phase 3 trials, thus sustaining fears that the vaccine approval process was driven by politics and not science [126]. 

Service providers (public health professionals, clinicians, researchers, policymakers etc.) also increase vaccine uptake in ACB communities. Our findings show that it is essential for providers to improve trust by acknowledging past and present racism, and seeking to understand the drivers of mistrust. Discussions about clients’ experiences of racism and bias are essential. They should include trauma-informed care, intersectionality, and reflections on provider bias with structural competency frameworks, given the sensitive nature of these subjects [127]. Findings also showed that a positive client-provider relationship could facilitate openness to education and recommendations about vaccines, decrease skepticism and increase vaccine confidence, improve care, and increase empowerment. Service providers must become more aware of social inequities to create interventions for social justice; healthcare equity occurs when no one is disadvantaged by their socially determined circumstances or position, and everyone has the opportunity to achieve their full potential [89,128]. Our findings show that vaccination discussions and healthcare provider recommendations impact ACB clients’ vaccination decisions; thus, service providers should not only provide accurate information but also understand the impacts of the SDOH, as well as advocate and implement interventions to counter disparities while focusing on helping those most vulnerable within the ACB community, such as those that are isolated, living alone, or depressed. Service providers must systematically and pro-actively work to advance health equity for Black people through strategies that address structural racism by building capacity, engaging in community-based participatory research, advocating and empowering vulnerable communities, and monitoring and evaluating interventions consistently over the long term [118]. Furthermore, they should also seek to understand factors that influence vaccination-related decisions and behaviors, support their clients in navigating the healthcare system, and remind them of the importance of vaccines. There is a moral imperative for service providers to address the health and socioeconomic needs of those who bear the intersecting burden of the structural discrimination that has led to inequities [129]. Healthcare providers must, therefore, seek to understand the needs of the ACB populations they serve, including having bidirectional and transparent conversations, counselling on specific vaccinations while addressing concerns, and having an awareness that building a trusting relationship is imperative, as some people such as Black mothers fear physicians will report their families to authorities. Pharmacist should also make efforts to be involved in the ACB community to help build confidence in vaccines. Black pharmacists, for example, recognize the structural and systemic issues that have contributed to the lower vaccination rates in Black communities. By collaborating with community leaders to disseminate evidenced-based information and by advocating and working to increase vaccine access, they can actively work to reduce vaccine inequities [66]. Furthermore, more effective therapeutic relationships and improved healthcare occur with racial concordance, which emphasizes the importance of racial diversity in healthcare as well as cultural humility [130]. Our results showed that ACB clients preferracially concordant professionals and find their messaging more impactful; therefore, efforts should be made to have more racially concordant professionals available for ACB clients. There is also a need for researchers to increase ACB participants in vaccine trials to build vaccine confidence in ACB populations. However, there was a gap in the literature regarding ways to increase ACB participants in vaccine trials as well as ways to increase racially concordant providers while being mindful not to increase their racial burden.

Engaging with ACB communities is an essential component of health promotion, and findings showed that culturally competent outreach, such as live stream town halls, can direct information to vulnerable populations while answering questions and countering misinformation to create transparency and help build trust. Through community collaborations, resources can be brought to the community where they are needed while helping to create community empowerment. Healthcare providers and organizations often have a relationship with community leaders to work towards creating interventions to meet the needs and preferences of the community [131]. In addition, by creating collaborative partnerships such as with faith-based organizations, educational committees, policy-makers, and researchers, joint efforts can be made to identify ways to improve the quality of care within ACB communities as well as create targeted messaging based on varied areas of expertise and experiences to address the needs of the community, including anti-racism initiatives and dispelling misconceptions. In order to create impactful, long-lasting change, these collaborations must be fostered and strengthened. 

Empowerment through engagement can help communities to become frontline vaccine advocates and mobilize and disseminate evidenced-based information, such as through faith-based community gatherings, providing testimonials about the benefits of vaccinations, and organizing community-based wellness programs. Community leaders and influencers should be strong advocates for ACB health, including being involved in all elements of the vaccine process, from its development to distribution. The church, for example, has social capital within ACB communities and can be a trusted source of information. Faith-based organizations can influence their members’ health behaviors on multiple socioeconomic levels; health initiatives that include cultural and spiritual contextualization have been shown to be effective [89]. Church-public health partnerships are needed to ensure that messaging is created and disseminated through a trusted voice in the Black community i.e., it stems from the church while reinforcing public health messaging [132]. 

Social interactions within ACB communities can also impact vaccine uptake. Family members were shown to impact the likelihood of vaccine acceptance; concerns of a family member could resonate through the family. It is also important to recognize that concerns about vaccines do not conflate into being vaccine resistant and that by addressing the questions and concerns of some, others in the family may also be impacted. However, a failure to address misinformation may further perpetuate and increase conspiracy theories and other concerns contributing to low vaccine uptake. Similarly, peers also impact the willingness to become vaccinated, based on the expectations of peers, wanting to protect others, and wanting to increase the ability to socialize at events together. The Peer Equity Navigator (PEN) program is an integrated knowledge mobilization initiative that uses the power of peers as experts on their community, culture and social connections to collaborate with researchers while undergoing critical racial literacy training to become community advocates and help build health equity in their communities [133]. Results also show that employing navigators and coordinators within the community can reduce accessibility issues; for example, through collaborations, they can establish community-based clinics. 

Social norms and context, such as the attitudes of social networks, can impact vaccine uptake; however, despite the perception that other Black people were vaccinated and perceived social approval within ACB communities both being shown to improve the willingness to be vaccinated, the social norms of the general population did not show this impact. In addition, the perception of communal safety through herd immunity may have created a false sense of protection and negatively impacted vaccine uptake. Results also showed that witnessing social leaders and service providers safely administer the vaccine may help build trust.

Another main theme in this review that pertained to low vaccine uptake in ACB communities was knowledge and communication, ranging from information, information sources, open dialogue, messaging, trusted messengers, message dissemination and interventions and strategies. The subtheme of information was further defined by lack of information, race-based information, and misinformation/myths/misconceptions/infodemic. Messaging ranged from diversity, clarity, acknowledgement, transparency, and more information, tailored to persuasive messaging. In addition, interventions and strategies included building trust, policies, addressing disparities, standing orders and protocols, and initiatives. 

Results showed that a lack of information is the main contributor to low vaccine uptake because it fostered uncertainty, because of unfamiliarity with the vaccine and the unknown. Information that was found to be generally lacking pertained to vaccine efficacy, safety, side-effects, outcomes, and lack of race-based information; these are important to make informed decisions. The ability to make informed decisions also impacts participation in clinical trials [134], which, as previously stated, low ACB participation in trials also contributed to low vaccine uptake in ACB communities. 

The reliability of the information is also paramount as misinformation, myths, misconceptions, and the infodemic cause low vaccine uptake in ACB communities; infodemic is a term referring to the rapid spread of both accurate and inaccurate information, coined by David Rothkopf in 2003 [135]. Misinformation campaigns were shown to target Black communities; susceptibility to misinformation is indicative of health literacy gaps in ACB communities, and this has been further exacerbated by the COVID-19-related infodemic. Organization health literacy is defined as “…the degree to which organizations equitably enable individuals to find, understand, and use information and services to inform health-related decisions and actions for themselves and others” [136] and it is interlinked with personal health literacy which is defined as “…the degree to which individuals have the ability to find, understand, and use information and services to inform health-related decisions and actions for themselves and others” [136]. More education is required to improve health literacy; however, there remains a gap in the literature about ways to improve health literacy in the ACB community to combat misinformation, targeted information with malicious intent, and the infodemic. 

Although organizations should establish and promote health literacy, they should also strive to be a trusted and well-used source of information. Results of this review showed that by identifying the most used, most trusted, and least trusted information sources, targeted public health campaigns can be created to improve vaccine uptake in ACB communities; for example, it should be considered that churches and virtual town halls were found to be good venues to disseminate information, while healthcare providers were found to be one of the most trusted sources of information and social media was found to be one of the least trusted sources of information. Results also showed that there was a higher level of VH in those that used less reliable sources and those who obtained information from health professionals had a better understanding. In addition, open dialogue with trusted professionals has been effective in helping to make informed choices. 

Messaging has been used by public health organizations to control the spread of disease by educating the public [137]. Messaging, both visual and narratives, must be racially diverse, clearly state the safety and efficacy of the vaccine, be comprehensive, and transparent, answer questions in a timely way, as well as acknowledge that racism, discrimination, and biases, past and present, have contributed to low vaccine uptake in ACB communities, which has caused mistrust; this must be recognized and acknowledged rather than placing blame on individuals by labelling them as VH. It must also be remembered that trust is part of the conversation but not all of it, as the reasons for low vaccine uptake in the ACB community are multifactorial.

The focus of interventions should be on education rather than becoming vaccinated; for example, there has been insufficient information in specific areas, such as vaccine efficacy and safety, to meet the general needs of ACB populations. It is also essential to provide information on resources, community support, and how becoming vaccinated can help to protect others. In addition, tailored messaging should be targeted to various groups within ACB communities; for example, based on age, to create more concise messages while providing evidence-based information and education through culturally appropriate and accessible messages. Culturally tailored messages must include the faces, voices, perspectives, and values of ACB audiences while addressing specific concerns, such as vaccination recommendations for those with comorbidities. Persuasive messaging can also be used to increase vaccine uptake and include narratives of people who have changed their minds and gotten vaccinated; messages that stress the benefits of vaccinating rather than the harm of not getting vaccinated were found to be more effective. Furthermore, to effectively disseminate messages, multiple venues must be used to increase access, including online and physical locations such as places of worship, barbershops, mobile clinics, and social events. 

In addition, to creating compelling messaging, trusted messengers could help improve vaccine uptake in ACB communities. Trusted messengers are considered to be trusted sources of health information by their community and can be trained to be ambassadors, and they can provide credible and culturally relevant information to those who may not seek information from healthcare authorities due to issues such as cultural and language barriers and low health literacy [138]. Results show that trusted messengers, such as pastors, Black healthcare providers, and faculty of Black students, are necessary for health communications by disseminating evidence-based information and providing vaccine promotion. They can also work with communities, governments, and institutions to build trust while endorsing vaccinations to help increase uptake. 

Several studies in this review introduced strategies and interventions to increase vaccine uptake in ACB communities, including ensuring participant protection for ethical research, screening for depression, and implementing a framework which involved understanding, listening, empathizing, engaging, and co-creating solutions. Furthermore, policies, such as making vaccinations mandatory, was shown to influence vaccine acceptance but also may have contributed to alienating others, such as parents who did not want their children vaccinated and contemplated removing them from school. Interventions must address disparities and work towards preventing the negative impacts of the SDOH; for example, racialized groups are more likely to be essential workers, use public transit, and more likely to live in dense housing areas, all of which increase exposure and vulnerability to COVID-19, it is through understanding critical racial literacy and analyzing structural inequities and systems of organizational power that the context of the lived experiences faced by racialized communities can be better understood [117]. In addition, although cost should not be a mitigating factor to receiving vaccines, monetary incentives were found to raise suspicions and were an ineffective strategy. By prioritizing populations that are more susceptible to infections, as the ACB community has shown with COVID-19, the larger population subsequently has increased protection through decreased risk of exposure. Initiatives should include sustained education about the benefits of vaccines, building trust and addressing the SDOH, improving access and health equity, addressing misinformation, and cultural competency training for service providers. 

Results also show that culturally responsive strategies must address the structural barriers associated with racism and everyday discrimination that create inequitable healthcare, including improving the ability to find and access services by streamlining services and having multiple medical and non-medical vaccination sites within communities, especially those in hard-to-reach areas. In addition, low quality and continuity of care are also associated with low vaccine uptake. Using standing orders and protocols for vaccinations, could increase vaccine availability and uptake. Furthermore, service providers should unify to increase mobile infrastructures with authentic communications, engagement, partnerships, and relevant messaging to increase health equity and remove barriers to vaccine uptake.

### 4.3. Micro/Individual

Through an intersectionality lens it is recognized that a person’s social identity is contingent on their life experiences and how they reconcile them through conscious and unconscious biases [139]. For example, religious and political beliefs were found to influenced vaccination beliefs; demonstrating how organizations and government systems (meso and macro) impact the individual. The variability in readiness, the dynamics of people’s views and access to health professionals based in the context of where they work and live must be considered when planning interventions [131]. 

The main theme of sentiments and behaviors helped to identify some of the factors associated with low vaccine uptake in ACB communities at a personal level and ranged from willingness to get vaccinated, vaccine beliefs, lived experiences, and vaccine confidence in our findings. In addition, willingness to get vaccinated further included subthemes of low necessity and increased willingness; vaccination beliefs included subthemes of general, religious, and political; lived experiences included the subthemes of mistreatment, death exposure, previous infection, previous vaccinations, and co-morbidities; and vaccine confidence included general distrust, the timeline and novelty of the vaccine, side effects, efficacy, and safety. 

Low willingness to get vaccinated was found to be impacted by the perception that other methods of protection such as personal protective equipment, staying home or taking supplements and being considered healthy made vaccination a low-level necessity. Others felt vaccinations where the wrong approach and that efforts should be made to improve baseline health. Doherty et al. [140], argues that adult vaccination is contingent of treating the choice to vaccinate as a public health issue, such as health diet, exercise, and smoking cessation, with a focus on relaying vaccination as a health-promoting activity rather than as a medical intervention against the spread of a specific pathogen. 

Factors that were shown to contribute to an increased willingness to get vaccinated included protecting themselves and others, that it was mandatory, to feel save and reduce worries, and to achieve a better sense of control of infection; skepticism about the vaccine was also associated with an increased willingness to get vaccinated and may be as a result of inquiry and obtaining more information. Therefore, it should be considered that vaccine mistrust that has been grounded in experiences of racism might not be a resistance to protective behaviors but rather a commitment to them [141]. In addition, vaccination beliefs influence vaccine uptake and are based on the individual’s perceptions of the benefits of the vaccine and the susceptibility and seriousness of an infection; those with who viewed the illness as more serious were more likely to be vaccinated. Doherty et al. [140], stated that two factors that are consistently associated with under-vaccination are low risk awareness of the vaccine-preventable disease and a poor understanding of the value that vaccination coverage in adults has. Vaccine confidence was negatively influenced by mistrust in the vaccine, personal fears, and misinformation; however, the literature also showed changes in VH which may indicate the willingness to become vaccinated can change of over time. Engaging with scientist, could foster a sense of responsibility, respect, and ownership of science, which could promote trust and has been associated with a greater perceived personal risk and a greater likelihood of taking preventative measures due to this risk [142]. Therefore, efforts should focus on addressing concerns highlighted in our results, including those related to the novelty of the COVID-19 virus, and the perceptions that the COVID-19 vaccine was created too quickly, as well as vaccine safety, side effects, and efficacy.

Results show that people’s lived experiences also influence their perceptions about vaccinations. For example, previous vaccinations were associated with vaccine receptivity and receiving the flu vaccine was found to be a predictor for receiving the COVID-19 vaccine; although, flu vaccination rates remain low in ACB populations. Those who personally experienced or witnessed a negative healthcare experience, and those who fear racial discrimination in their healthcare, were also less likely to be vaccinated. Furthermore, having a friend or family member die was found to be associated with a higher rate of VH; however, a causal link was not studied, and it was not known if those who died were also considered VH. In addition, having a previous COVID-19 infection was not largely associated with the willingness to get vaccinated, even if the vaccine was safe and effective. It was also shown that some with comorbidities were less likely to get vaccinated for fear that it would strain their immune system or cause fare-ups in their illness/ disease. Sternber [143] stated that lack of information of vaccine clinical trials involving participants with comorbidities maybe a reason for the apparent VH. However, those living with HIV generally had higher levels of COVID-19 vaccination, particularly in those who perceived the illness as dangerous. Padamsee et al. [141], found that Black individual’s belief that the COVID-19 vaccination was necessary for protection increased more rapidly than White people, however, vaccination rates continuing to be lower demonstrated that vaccine sentiments are not the sole factors contributing to low vaccine uptake in ACB communities. 

Race, socioeconomic status, and health burdens are some of the factors that have contributed to ACB populations’ vulnerabilities [144], as past experiences, and social structures impact behavior [114]. This ScR has sought to further understand the complexities of low vaccine uptake of ACB populations in high income countries by identifying and defining the main concepts and their boundaries in the literature. The main concepts of inequities and racism, sentiments and behaviors, knowledge and communication, and engagement and influence, were identified and defined across the macro, meso, and micro levels. These concepts were further defined through an intersectionality lens and with consideration to the social determinants of health; as social and structural determinants of health permeate throughout where ACB populations work, live, learn, and pray [145]. By defining the boundaries of these concepts gaps in the literature were highlighted. These gaps warrant further investigations to improve the understanding of low vaccine uptake in ACB populations. These knowledge gaps include knowledge about how historical racism can be effectively addressed in a manner that can build/rebuild trust in our current healthcare system; defining why racism and discrimination in the healthcare system still currently exists; how to support racially concordant professionals, particularly their mental wellbeing; how to support the responsibilities of effective vaccination communication to build a trusting relationship during times of dynamic changed; effective methods to increase ACB participants in vaccine trials or methods to increase racially concordant providers to serve ACB clients; and how to improve health literacy in the ACB community to combat misinformation, targeted information with malicious intent and the infodemic. 

## 5. Conclusions

The World Health Organization [13] defines vaccine hesitancy as the refusal or reluctance to become vaccinated despite available vaccines; reasons identified include complacency, inconvenience, and lack of confidence, this fails to capture the mistrust from historic and current racism and discrimination including the lack of cultural competencies in healthcare practice that could build rapport and build trust. This study has helped to identify and define the problem of low vaccine uptake in ACB populations and has demonstrated the breath of the issues, as vaccine access is not only physical barrier but also a knowledge, psychological, and socioeconomic barrier at the individual, organizational, and systemic level to form structural inequities that have manifested as low vaccine uptake.

### Strengths and Limitations 

This scoping review is strengthened by the anticipated broad selection and types of evidence sources, including grey literature, to maximize the breadth of findings. However, this study is limited by the dynamic nature of vaccine hesitancy, which will require surveillance and acquiring of new information to remain current. The restriction of language to English and French is another limitation, as there is the potential to lose valuable data. Another limitation is that the evidence sources were not critically appraised, which impacts bias and the validation of this study, as well as the ability to synthesize and analyse the extracted findings. In addition, this ScR is limited by the exclusion of statistical meta-synthesis, which impedes the ability to make recommendations for healthcare related practices and interventions that may facilitate vaccine uptake in ACB and other vulnerable populations. Furthermore, due to the barrier of lack of race-based data in Canada and some other parts of the world, our literature search mostly consisted of studies from the US thus impacting the generalizability of the studies findings.

## Figures and Tables

**Figure 1 vaccines-12-00269-f001:**
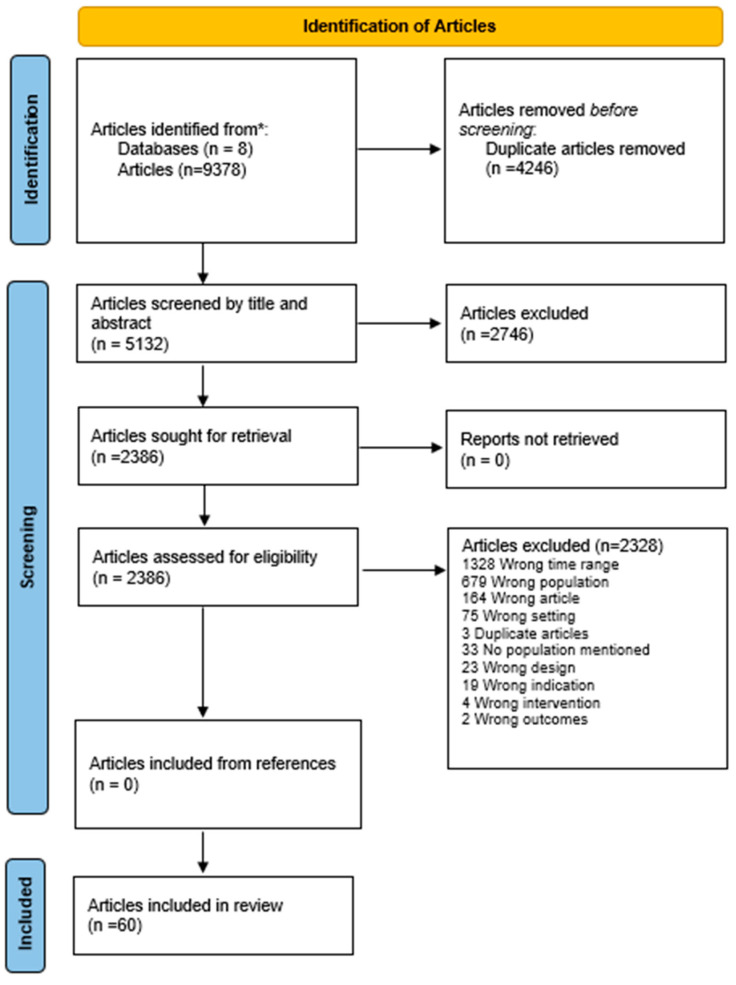
Prisma diagram. * This Prisma diagram shows the selection process from the initial retrieved articles from 8 databases to the resultant 60 articles selected to incorporate into this scoping review, with reasons for articles not being included [50].

## Data Availability

Dissemination will occur through peer-reviewed open-access journals and conferences that target stakeholders in public health, vaccination campaigns and overcoming inequities in healthcare.

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
