# Peer review of "Understanding Low Vaccine Uptake in the Context of Public Health in High-Income Countries: A Scoping Review"

_vaccines, 2024, doi:10.3390/vaccines12030269_

Round 1

Reviewer 1 Report

Comments and Suggestions for Authors

Dear author,

Greetings!

1 Please check similarity publication  (By using turnit in soft ware )

2Understanding Low Vaccine Uptake in African, Caribbean, and 2 Black Populations Relative to Public Health in High-Income 3 Countries: A Scoping Review ( Please modify title )

3 This scoping review aims to investigate low vaccine uptake in ACB 20 populations relative to public health in high-income countries. A search was conducted in: MED- 21 LINE(R) ALL (OvidSP), Embase (OvidSP), CINAHL (EBSCOHost), APA PsycInfo (OvidSP), 22 Cochrane Central Register of Controlled Trials (OvidSP), Cochrane Database of Systematic Reviews 23 (OvidSP), Allied & Complimentary Medicine Database (Ovid SP), and Web of Science following the 24 Joanna Briggs Institute (JBI) framework for scoping reviews, supplemented by PRISMA-ScR. Theo- 25 retical underpinnings of the intersectionality approach were also used to help interpret the com- 26 plexities of health inequities in the ACB population. (Please verify journals and cite their references instead of websites and sites)

4Introduction 40 African, Caribbean, and Black (ACB) populations are not only vulnerable due to 41 health inequities, as evidenced by higher rates of SARS- CoV-2 infections, hospitaliza- 42 tions, and associated mortalities,(Please justify and follow author and journal standard guidelines )

5In Canada and globally, vaccines significantly prevent and control infectious diseases 77 and are thus a cornerstone of public health [21]. Historically vaccines have reduced dis- 78 ease-specific mortality rates including smallpox, rabies, polio, the plague, typhoid and 79 many more, and have significantly decreased infant mortality rates globally [18]; over 3 80 million child deaths are estimated to be prevented each year globally through vaccina- 81 tions [22]. Despite vaccinations being considered to be one of the public health’s greatest 82 success stories, vaccine hesitancy is influenced by the confidence in the competencies of 83 health professionals and health services [23,24]. Vigilance is required to maintain and in- 84 crease vaccine uptake, especially in vulnerable populations. People’s behaviors and will- 85 ingness to follow recommended measures are the most powerful tools ag with ainst the viral 86 spread [25,26]. (Please check citation and references with authentification)

ACB populations in the JBI Evidence Synthesis, Cochrane Database of Systematic Re- 112 views, Cumulative Index to Nursing and Allied Health Literature (CINAHL) and Pub- 113 Med. Several terms were piloted (refer to Appendix A for the search strategy), and the 114 keywords “vaccine hesitancy “ and “Black” yielded the most relevant results, namely, the 115 following four reviews: One rapid systematic review related to COVID-19 vaccine hesi- 116 tancy and minority ethnic groups in the UK [34]; one scoping review related to COVID- 117 19 vaccine hesitancy globally [35]; and two systematic reviews, one related to vaccine hes- 118 itancy in the US [36] and the second related to vaccine acceptance in different populations 119 in the US [37].(Please modify and check review of literature )

To determine existing interventions to improve low vaccine uptake in the study 138 population. These objectives were achieved through systematically reviewing the breadth 139 and types of source evidence available; the software program Covidence was used to man- 140 age and select source evidence, which was open to all types of evidence sources, and data 141 extraction from source information was done manually. If the data identified ways that 142 public health addresses or fails to address vaccine uptake, these were reported in the find- 143 ings through descriptive narrative.(Please modify this major correction including gender ,age etc)

Intersectionality, the socioeconomic model (SEM), and the social determinants of 150 health (SDOH) approach will be used to help interpret the complexities related to low 151 vaccine uptake in ACB populations. Intersectionality highlights the unique forms of 152 Vaccines 2023, 11, x FOR PEER REVIEW 4 of 73 discrimination faced by ACB and other vulnerable populations. Through this perspective, 153 it is recognized that inequities result from multiple factors related to the intersection of 154 power relation, experiences, and social locations [39]; the intersection of these factors ex- 155 acerbates health inequities [40]. In addition, the socioeconomic model (SEM) describes 156 how characteristics of the environment and the individual interact at multiple levels 157 (macro/systems, meso/organizational, and micro/individual) to influence health out- 158 comes [41].(Please justify and revise )

references and figures follow journal format 

check plagiarism 

Best regards, 

Comments on the Quality of English Language

minor revision is required mendeley grammerley works

Author Response

Manuscript submitted to Vaccines Journal

Previous title: Understanding Low Vaccine Uptake in African, Caribbean, and Black Populations Relative to Public Health in High-Income Countries: A Scoping Review

Response to Reviewers comments, February 2nd, 2024

Item #

Reviewers Comments (Reviewer 1)

Authors’ Responses

*Revisions have been highlighted in the manuscript

1.       

Please check similarity publication

We have carefully reviewed the manuscript and the only similarities is with the protocol of the scoping review that has not yet been published but submitted to preprint.

2.       

Please modify title (line 2-3)

Thank you for this feedback, the title has been modified to reflect the scope of the study.

“Understanding low vaccine uptake in a context of public health in high-income countries: A scoping review.”

Article Type: Scoping Review

3.       

Please verify journals and cite their references instead of websites and sites (line 20-26)

All journals and citations have been verified, now websites and sites have been stated. Also, no references required cited within abstract (i.e. line 20-26) section of the paper.

A search was conducted in: MEDLINE(R) ALL (OvidSP), Embase (OvidSP), CINAHL (EBSCO-Host), APA PsycInfo (OvidSP), Cochrane Central Register of Controlled Trials (OvidSP), Cochrane Database of Systematic Reviews (OvidSP), Allied & Complimentary Medicine Database (Ovid SP), and Web of Science following the Joanna Briggs Institute (JBI) framework for scoping reviews, supplemented by PRISMA-ScR. Theoretical underpinnings of the intersectionality approach were also used to help interpret the complexities of health inequities in the ACB population. The eligibility criteria were based on the Population.

-        As there are no websites in the text above

4.       

Please justify and follow author and journal standard guidelines (line 40-42)

We have reviewed and followed guidelines that were provided, including using the Vaccines Journal template document.

5.       

Please check citation and references with authentication (line 76-86)

After reviewing citations and the corresponding references in line 76-86, we confirmed that the in-text citation and corresponding references are aligned and are relevant to the contents of the manuscript.

6.       

Please modify and check review of literature (line 111-119)

We have reviewed and checked the specified text within lines 111-119 and we confirmed that they address the preliminary search that was done in the study. We confirmed that the in-text citation and corresponding references are aligned and are relevant to the contents of the manuscript.

7.       

Please modify this major correction including gender, age etc (line 144-150)

There seems to be a misalignment with this comment and the specified text identified, as there is no mention of gender or age within those highlighted lines.

8.       

Please justify and revise (line 154-163)

Section has been reworded to improve clarity.

Reviewer 2 Report

Comments and Suggestions for Authors

This paper describes an interesting and important issue, namely vaccine hesitancy and potential disproportional (low) uptake of vaccines in special population groups. The scoping review is very comprehensive and methodologically sound. However, the presentation is excessive and overly long and detailed, missing the main point of a review, namely to summarise a large amount of information and data into a single, simple and easily readable text with clear conclusions. It is recommended that the text is significantly reduced.  

The paper fails to acknowledge and address some important shortcomings. First of all, almost all of the studies in the review were done in the US (with a few from Canada and UK). This makes it very questionable whether the findings and conclusions can be generalised to other locations.  The title indicates "High-income countries" but in reality it is only US. This should be made clear. 

The review aims to understand low vaccine uptake in ACB populations, but does not adequately include a proper comparator, for example Asian or Hispanic populations. This would be interesting to include, or at least reflect in the Discussion. It is not clear whether the conclusions in the paper are specific for ACB or apply to other minorities as well.

Similarly, it is not clear whether similar barriers to vaccine uptake as revealed in the paper can be found generally in low socio-economic groups. In other words, the the low uptake of vaccines in ACB could be linked to low socio-economic status rather than to race. This would be relevant to include in discussion or acknowledge as a limitation of the study.    

Author Response

Manuscript submitted to Vaccines Journal

Previous title: Understanding Low Vaccine Uptake in African, Caribbean, and Black Populations Relative to Public Health in High-Income Countries: A Scoping Review

Response to Reviewers comments, February 2nd, 2024

Item #

Reviewers Comments (Reviewer 2)

Authors’ Responses

*Revisions have been highlighted in yellow in the manuscript

1.       

This paper describes an interesting and important issue, namely vaccine hesitancy and potential disproportional (low) uptake of vaccines in special population groups. The scoping review is very comprehensive and methodologically sound. However, the presentation is excessive and overly long and detailed, missing the main point of a review, namely to summarise a large amount of information and data into a single, simple and easily readable text with clear conclusions. It is recommended that the text is significantly reduced.  

Thank you for this feedback, we have reviewed the entire manuscript and reworded sentences to improve clarity and shorten the length of manuscript. However, as in any scoping review, the length of the table has taken 50% of this paper which accounts for a significant part of the length.

2.       

The paper fails to acknowledge and address some important shortcomings. First of all, almost all of the studies in the review were done in the US (with a few from Canada and UK). This makes it very questionable whether the findings and conclusions can be generalised to other locations.  The title indicates "High-income countries" but in reality, it is only US. This should be made clear. 

We acknowledge the limitations of the study including the lack of representation of all high-income countries. Below are the steps we have taken to address this concern:

We have included a statement on page 3, paragraph 3, line 132-139, highlighted in yellow explaining the reason for the disproportionately high number of studies from the US addressing ACB health. It states:

It is imperative to note that there is a dearth of research within the literature that adequately address the challenges that ACB populations face with vaccination programs. While this scoping review include all high-income countries, the articles that met the inclusion criteria were mostly from the US, Canada and the UK…”

We have also included a statement that addresses the limitations of having mostly US studies in the last sentence in the limitations section on page 61, line 1737-1739. It is highlighted in yellow, and it states:

Furthermore, due to the barrier of lack of race-based data in Canada and some other parts of the world, our literature search mostly consisted of studies from the US thus impacting the generalizability of the studies findings”.

3.       

The review aims to understand low vaccine uptake in ACB populations, but does not adequately include a proper comparator, for example Asian or Hispanic populations. This would be interesting to include, or at least reflect in the Discussion. It is not clear whether the conclusions in the paper are specific for ACB or apply to other minorities as well.

This paper is the result of a scoping review and majorly focused on ACB people as we have observed that they are a vulnerable population in Canada and had a high level of infection rate. You can find this highlighted in yellow on page 1 (line 40-43) and page 2 (line 49-53, 1st paragraph).

They state: “African, Caribbean, and Black (ACB) populations are not only vulnerable due to health inequities, as evidenced by higher rates of SARS- CoV-2 infections, hospitalizations, and associated mortalities, but are also the least willing to receive the coronavirus disease of 2019 (COVID-19) vaccine” and “Although race-based data collection remains inconsistent in Canada, the cities of Ot-tawa and Toronto reported 1.5-5 times the increase in COVID-19 infection rates among racialized communities; these findings are consistent with other high-income countries, including the United States (US) and the United Kingdom (UK).”

-        Because the aim of a scoping review differs from that of a systematic review, question development may not fit into the PICO (Patient/Intervention/Comparison/Outcome) framework. Therefore, PCC (Population/Concept/Context) may be a more useful framework. (See details in JBI Scoping Review Guideline)

To ensure that the paper would mainly focus on the issues related to ACB people we selected only the evidence sources that had a primary focus on African, Caribbean, and/or Black populations. You can see this highlighted in yellow on page 5 (line 239-241) and further throughout the methods section.

It states: “…only evidence sources with a primary focus on African, Caribbean, and/or Black populations will be included so that issues that are specific to this target population are ad-dressed.”

4.       

Similarly, it is not clear whether similar barriers to vaccine uptake as revealed in the paper can be found generally in low socio-economic groups. In other words, the low uptake of vaccines in ACB could be linked to low socio-economic status rather than to race. This would be relevant to include in discussion or acknowledge as a limitation of the study.    

While we understand that some of the factors contributing to low-vaccine uptake may be common across different marginalized groups, the focus of this scoping review was on ACB population. From our scope of the literature, we observed 4 major themes that could be contributing to low vaccine uptake in ACB people. Within these 4 themes we have highlighted how socioeconomic status plays a role. You can see these highlighted in blue on pages, 35 (line 429-432), 36 (line 451-454), 53 (line 1303-1307). As we use the intersectionality framework to perform our studies, we show how these various factors intersect to impact uptake not focusing on one factor alone.